_Article_

# Structural basis for EtfD-mediated coupling of β-oxidation and the respiratory chain in mycobacteria

Gautier M Courbon [ID] [1,2 ✉], Vadim Makarov [ID] [3], Stewart T Cole [ID] [4], Dirk Schnapinger [ID] [5], Sabine Ehrt [ID] [5] & John L Rubinstein [ID] [1,2,6 ✉]

## Abstract

Targeting β-oxidation has been proposed as a strategy for shortening tuberculosis (TB) treatment by killing non-replicating _Mycobacterium tuberculosis_ within granulomas where the pathogen relies on host-derived lipids. The protein EtfD is thought to couple β-oxidation of fatty acids with the respiratory chain in mycobacteria. However, the structure of EtfD is not known and, as the presumed link between two complex processes, its activity has been difficult to measure, impeding its exploitation as a drug target. Here we show that _Mycobacterium smegmatis_, a fast growing and nonpathogenic model for _M. tuberculosis_, relies on EtfD for extracting energy from β-oxidation. The electron cryomicroscopy structure of _M. smegmatis_ EtfD reveals an unusual linear [3Fe-4S] cluster that has not been seen in other protein structures, and suggests how EtfD transfers electrons from β-oxidation to the respiratory chain. We devised an assay that couples EtfD activity to a fluorescent readout of proton pumping by the respiratory chain, which can be used to identify compounds that block mycobacteria from using β-oxidation to power oxidative phosphorylation.

**Keywords** ETF-QO; EtfD; Beta Oxidation; Mycobacteria; Structure
**Subject Categories** Metabolism; Microbiology, Virology & Host Pathogen Interaction; Structural Biology

## Introduction

During infection, _Mycobacterium tuberculosis_, which causes the disease tuberculosis (TB), can survive within structures in the lung formed by immune cells and known as granulomas (Russell, 2001; Barry et al, 2009). In the lipid-rich and hypoxic environment of the granuloma, the bacterium undergoes metabolic remodeling that allows it to enter a persistent non-replicating state associated with tolerance to drugs, necessitating long treatment durations for infections (Gomez and McKinney, 2004; Boshoff and Barry, 2005;

Ehrt et al, 2018). The discovery and clinical success of the adenosine triphosphate (ATP) synthase inhibitor bedaquiline, which kills both replicating and non-replicating _M. tuberculosis_ in vitro (Koul et al, 2008), has validated mycobacterial energy metabolism as a target for the development of new drugs with the potential to shorten TB treatment (Andries et al, 2005; Deshkar et al, 2022). The use of cholesterol and fatty acids as a carbon source in granulomas, where other nutrients are scarce, could play an important role in persistence (Boshoff and Barry, 2005; Russell et al, 2009). Therefore, disrupting _M. tuberculosis_ lipid catabolism has been proposed as a strategy to target persistent mycobacteria, which could also reduce treatment duration (Wilburn et al, 2018; Beites et al, 2021).

Metabolism of fatty acids to release energy uses the β-oxidation pathway. This pathway is tightly integrated with the central carbon metabolism of the Krebs cycle (Fig. 1A). In β-oxidation, fatty acids are activated by an acyl-CoA synthetase that attaches coenzyme A (CoA) to the first carbon of the acid, forming a fatty acyl-CoA. The fatty acyl-CoA is subjected to multiple iterations of processing by acyl-CoA dehydrogenases, enoyl-CoA hydratases, β-hydroxyacyl-CoA dehydrogenases, and thiolases. Each iteration shortens the fatty acid's aliphatic chain by two carbons, releases a molecule of acetyl-CoA, reduces a soluble nicotinamide-adenine dinucleotide molecule ($NAD^+$) with two electrons to form NADH, and reduces a protein-bound flavin adenine dinucleotide (FAD) cofactor with two electrons to form $FADH_2$. The NADH and $FADH_2$ are oxidized by the respiratory chain or electron transport chain (ETC), which ultimately establishes an electrochemical proton motive force (pmf) across the plasma membrane. The pmf is used by ATP synthase to generate ATP. The acetyl-CoA from β-oxidation enters the Krebs cycle, leading to the reduction of either two $NAD^+$ molecule and one protein-bound FAD, or one $NAD^+$ molecules and two protein-bound FADs, depending on which malate oxidizing enzyme is used by the organism (Houten and Wanders, 2010; Harold et al, 2022).

The FAD cofactor reduced in β-oxidation is found in a FadE protein, an acyl-CoA dehydrogenase, which transfers electrons to a FAD-containing Electron Transfer Flavoprotein (ETF). Although FAD can be reduced by two electrons to form $FADH_2$, ETF has been proposed to oxidize acyl-CoA dehydrogenases in two successive one-electron steps requiring two ETF (Hall and

[1] Molecular Medicine Program, The Hospital for Sick Children, Toronto, Ontario, Canada. [2] Department of Medical Biophysics, The University of Toronto, Toronto, Ontario, Canada. [3] Research Center of Biotechnology, Russian Academy of Sciences, Moscow, Russia. [4] Ineos Oxford Institute for Antimicrobial Research, University of Oxford, Oxford, UK. [5] Department of Microbiology and Immunology, Weill Cornell Medicine, New York, NY, USA. [6] Department of Biochemistry, The University of Toronto, Toronto, Ontario, Canada. ✉ E-mail: gautier.courbon@mail.utoronto.ca; john.rubinstein@utoronto.ca

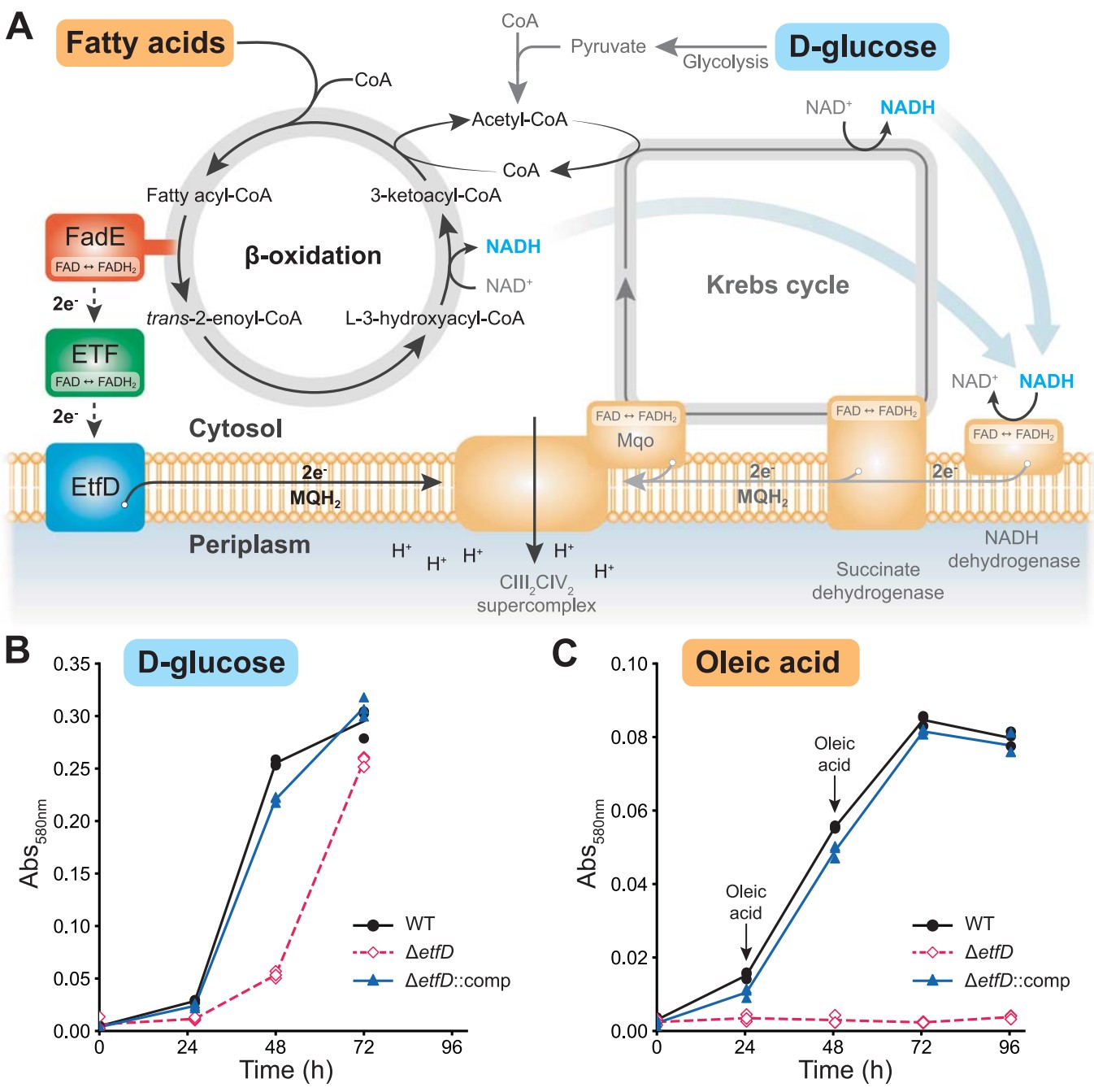

**Figure 1.   M. smegmatis EtfD is essential for growth on fatty acids.**

(A) Schematic representation of the Krebs cycle and β-oxidation in *M. smegmatis*. (B) Growth of *M. smegmatis* on modified Sauton's medium supplemented with 11 mM D-glucose. EtfD-3×FLAG (GMC_MSM4) is used as the wild-type (WT) control. (C) Growth of *M. smegmatis* on modified Sauton's medium supplemented with 250 μM oleic acid. Each arrow indicates addition of 250 μM oleic acid. Each point shows one of three replicates and the plot is representative of two independent experiments. Source data are available online for this figure.

Lambeth, 1980; Gorelick et al, 1985). Mycobacterial ETF (encoded by *rv3028c* and *rv3029c*, also named *etfAB*) and mammalian ETF are homologous. In mammals, an ETF:quinone oxidoreductase (ETF-QO, encoded by the gene *ETFDH*) transfers electrons from ETF to ubiquinone in the mitochondrial inner membrane, which reduces complex III (also known as cytochrome *bc*₁) of the ETC. Consequently, mitochondrial ETF and ETF-QO link β-oxidation to

the ETC, just as succinate dehydrogenase (Sdh, also known as complex II) (Hägerhäll, 1997) links the Krebs cycle to the ETC. The genes *rv0338c* in *M. tuberculosis* and *MSMEG_0690* in *Mycobacterium smegmatis*, a fast-growing model for mycobacterial energy metabolism, encode a protein named EtfD that has been proposed to serve as an ETF-QO in these organisms (Wischgoll et al, 2005; Agne et al, 2021; Beites et al, 2021). EtfD is expected to transfer

electrons from ETF to menaquinone in the plasma membrane, reducing it to menaquinol, which transfers electrons to the complex III$_2$IV$_2$ supercomplex (cyt. *bcc-aa*$_3$) or cyt. *bd* (Beites et al, 2021; Zhang et al, 2006). However, mycobacterial EtfD is not homologous with mammalian ETF-QO.

The *M. tuberculosis* genome encodes many different enzymes that could participate in β-oxidation, including 35 putative acyl-CoA dehydrogenases (Beites et al, 2021; Cole et al, 1998). This diversity leads to an apparent functional redundancy that would make it difficult to target *M. tuberculosis* β-oxidation with inhibitors. However, mycobacterial genomes encode a single ETF and a single EtfD, suggesting a vulnerability in fatty acid catabolism that could be susceptible to inhibition by a single small molecule (Beites et al, 2021). Deletion of EtfD prevents growth of *M. tuberculosis* on fatty acids and cholesterol in vitro, and prevents infection in mouse models of TB (Beites et al, 2021). Further, a series of compounds have been shown to rely on EtfD to kill *M. tuberculosis* (Székely et al, 2020). The essential nature of EtfD and its lack of homology to human ETF-QO make it an attractive target for treatment of TB. However, the absence of an experimental structure of the protein and the absence of an assay for its activity limit identification and optimization of inhibitors.

In this study, we show that, as in *M. tuberculosis*, deletion of EtfD prevents *M. smegmatis* growth when fatty acids are the sole carbon source available, supporting the use of *M. smegmatis* in the characterization of EtfD inhibitors. We used electron cryomicroscopy (cryo-EM) to determine the *M. smegmatis* EtfD structure, revealing unusual iron-sulfur clusters including a linear [3Fe-4S] cluster that had been produced synthetically and observed spectroscopically but had not been resolved in a protein structure. The iron-sulfur clusters and a *b* heme form a wire that allows electron transfer from ETF in the mycobacterial cytosol to menaquinone in the plasma membrane. Finally, we develop an in vitro assay for EtfD activity. The assay confirms the protein's role in linking fatty acid-oxidation to proton pumping by the ETC and will allow for direct measurement of inhibition of its activity, potentially in high-throughput screening for drug discovery.

## Results

### EtfD is needed for *M. smegmatis* growth with fatty acids as the sole energy source

Deletion of EtfD in *M. tuberculosis* (*rv0338c*) prevents growth in medium that contains fatty acids or cholesterol as the sole carbon source (Beites et al, 2021). This observation suggests that a phenotypic screen for growth on medium containing fatty acids may be useful for detecting EtfD inhibitors. However, *M. tuberculosis* has a doubling time of 24 h and is a human pathogen, complicating its use in high-throughput screens. In contrast, *M. smegmatis* has a doubling time of 3 to 4 h and is nonpathogenic (Reyrat and Kahn, 2001). The value of *M. smegmatis* as a model for energy metabolism in mycobacteria is supported by the phenotypic screen that led to the development of bedaquiline (Andries et al, 2005). Further, *M. tuberculosis rv0338c* and *M. smegmatis MSMEG_0690* are closely related, with 80.2% sequence similarity and 71.2% sequence identity for the folded region of the protein (*MSMEG_0690* residues 1–779, *rv0338c* residues 1–742). For the

predicted disordered C-terminal region, the sequence identity and similarity are 36.9% and 39.9%, respectively, with 46.8% gaps. Despite this similarity, whether or not *M. smegmatis* also relies on EtfD for growth on fatty acids is not known. Therefore, we deleted the gene encoding EtfD (*MSMEG_0690*) from *M. smegmatis* using the ORBIT method (Murphy et al, 2018) and tested the ability of the Δ*etfD* strain to grow in liquid medium containing either glucose or oleic acid as the sole carbon source (Fig. 1B,C, from *n* = 3 replicates and representative of two independent experiments). With glucose as the sole carbon source, which provides energy through glycolysis and the Krebs cycle, the Δ*etfD* strain demonstrates a slight growth delay (Fig. 1B, *pink*) compared to a wild-type strain (Fig. 1B, *black*). A similar delay was reported previously for *M. tuberculosis* Δ*etfD* when grown on glycerol (Beites et al, 2021). The growth defect was rescued by complementing the Δ*etfD* strain with a plasmid that allowed expression of *M. tuberculosis* EtfD (Fig. 1B, *blue*). With oleic acid as the sole carbon source, which provides energy through β-oxidation, Δ*etfD M. smegmatis* did not grow (Fig. 1C, *pink*), whereas the wild-type strain displayed robust growth (Fig. 1C, *black*). Again, this growth defect was rescued by complementation with a plasmid that allowed expression of *M. tuberculosis* EtfD (Fig. 1C, *blue*). These data demonstrate that, like *M. tuberculosis*, *M. smegmatis* requires EtfD for growth on fatty acids, and that expression of *M. tuberculosis* EtfD rescues deletion of *M. smegmatis* EtfD. This similarity suggests that *M. smegmatis* may be used in phenotypic screens and characterization of inhibitors of *M. tuberculosis* EtfD.

### Structure of EtfD reveals an electron wire that connects ETF to the menaquinone pool

EtfD has been proposed to transfer electrons from soluble ETF to the membrane-bound menaquinone pool, analogous to human ETF-QO (Beites et al, 2021). However, the two proteins are not homologous and the structural basis for this activity in mycobacteria remains unclear. To investigate this process, we prepared a strain of *M. smegmatis* with a 3×FLAG tag at the C terminus of EtfD, cultured the cells, and isolated membranes. Membranes were then solubilized with the detergent dodecyl maltoside (DDM) and EtfD was purified by affinity chromatography. Cryo-EM of the sample yielded a three-dimensional (3D) map of EtfD at 3.2 Å resolution (Fig. 2A, *left*, Fig. EV1, Table EV1), with resolution in the soluble region reaching 2.8 Å (Fig. EV2). This resolution enabled construction of an atomic model for 71% of the 1042-residue protein (Fig. 2A, *right*, Table EV1). The missing residues in the model correspond to the initial methionine, two loops from residues 348–373 and 621–628, and the ~30 kDa (residues 778–1042) disordered region at the C terminus of the protein. The ordered portion of EtfD is mostly α-helical (Fig. 2A, *right*), consisting of a membrane region comprising five transmembrane α helices and a soluble region that contains two [4Fe-4S] binding domains and two cysteine-rich CCG domains (Fig. 2B). The two CCG domains coordinate a linear [3Fe-4S] cluster (cluster D1) and a noncubane [4Fe-4S] cluster (cluster D2) (Fig. 2C, Movie EV1). The linear [3Fe-4S] cluster D1 is coordinated at each end by Cys503, 538, 584, and 581 (Fig. 2D, *top left*). The cluster consists of two perpendicular diamond-shaped planes, matching the structure reported in a chemically synthesized linear [Fe$_3$S$_4$(SPh)$_4$]$^{3-}$ cluster (Hagen and Holm, 1982). The noncubane [4Fe-4S] D2 cluster is coordinated by Cys637, 672, 673, 709, 712, as well as His583 (Fig. 2D,

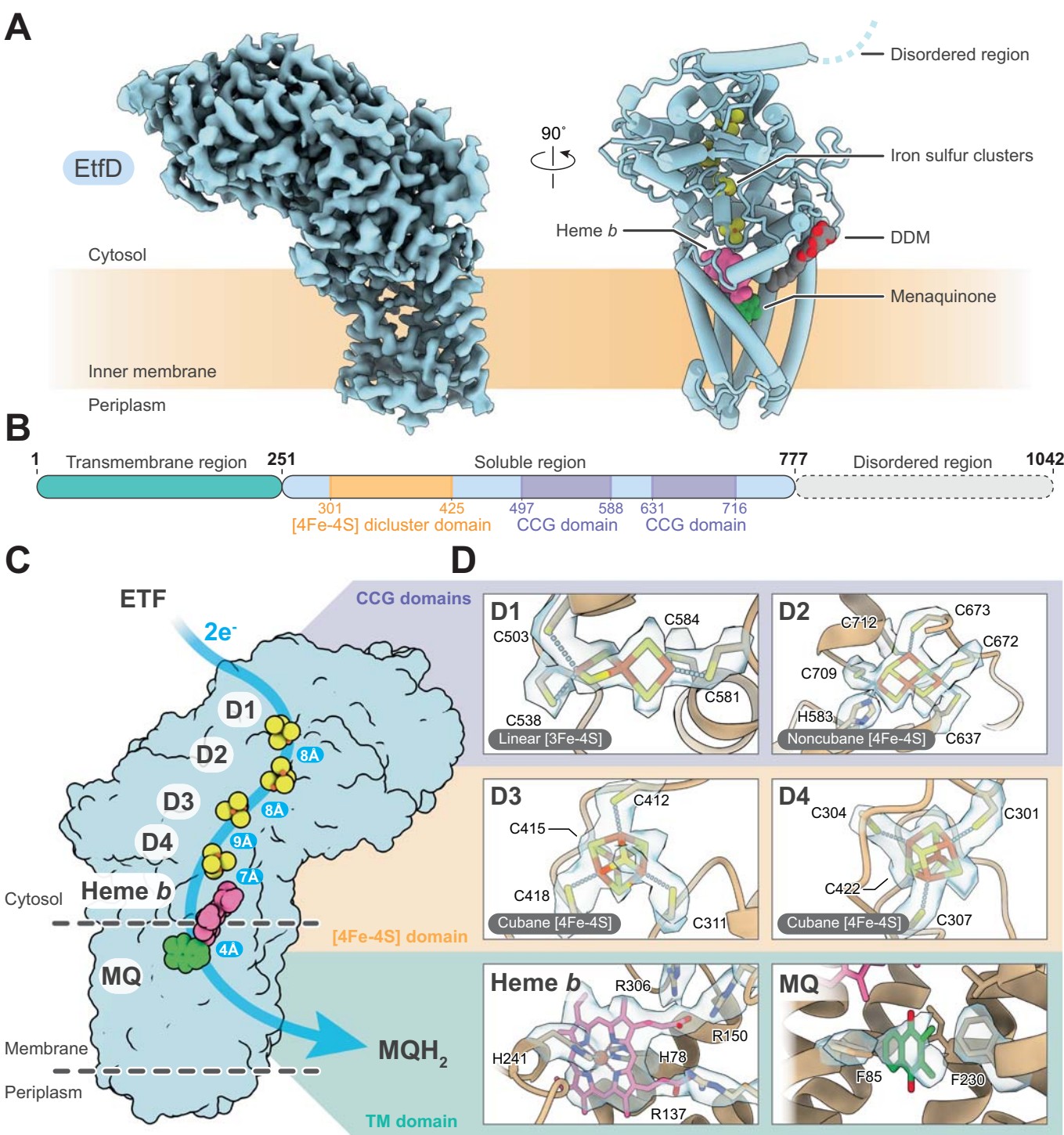

**Figure 2. Cryo-EM structure of *M. smegmatis* EtfD.**

(A) Cryo-EM map (*left*) and atomic model (*right*) of EtfD. DDM, dodecyl maltoside. (B) Schematic representation of EtfD domain organization. Domains were annotated using Pfam (Finn et al, 2008). (C) Arrangement of redox cofactors in EtfD. Iron-sulfur clusters are shown in yellow and orange, heme *b* in pink, and menaquinone in green. All cofactors are shown as space filling models. A possible electron path from ETF to menaquinone is shown by the blue arrow and the edge-to-edge distance between redox cofactors is indicated. (D) Coordination of the redox cofactors in EtfD. The cryo-EM map density is shown in blue.

*top right*). These unusual iron-sulfur clusters are discussed in detail in the next section. The two [4Fe-4S] domains coordinate two cubane [4Fe-4S] clusters (clusters D3 and D4) (Fig. 2C). Cluster D3 is coordinated by Cys311, 412, 415, and 418 while cluster D4 is coordinated by Cys301, 304, 307, and 422 (Fig. 2D, *middle*). The membrane-embedded region of EtfD contains a heme that, based on the map density and coordinating residues, appears to be a *b* heme (Fig. 2D, *bottom left*). His78 and 241 coordinate the metal center of the heme, with Arg137, 150, and 306 forming ionic interactions with the heme's carboxy groups. Additional density in the map 4–5 Å from the edge of the heme accommodates what appears to be the napthoquinone head of a menaquinone molecule and likely corresponds to the menaquinone binding site (Fig. 2D, *bottom right*). The interaction of this putative menaquinone with the protein is stabilized by aromatic interactions with Phe85 and Phe230. The membrane region of EtfD also appears to interact stably with a detergent-like density that we assigned to a DDM molecule (Fig. 2A, *right*).

The iron-sulfur clusters, heme, and menaquinone in EtfD are sufficiently close to allow rapid electron transfer between them, with cluster D1 located ~8 Å away from cluster D2, cluster D2 ~8 Å from cluster D3, cluster D3 ~9 Å from cluster D4, cluster D4 ~7 Å from heme *b*, and heme *b* ~4 to 5 Å from the putative menaquinone binding site (Fig. 2C). Therefore, the cryo-EM map suggests that EtfD forms a continuous electron wire that can link ETF in the cytosol to the menaquinone pool in the membrane. In this process, electrons could flow sequentially from the FAD in ETF to cluster D1, D2, D3, D4 and the *b* heme, which would reduce menaquinone to menaquinol. However, without knowing where ETF binds on the surface of EtfD, it is not clear which cluster mediates the initial entry of electrons from the FAD of ETF into EtfD.

## Unusual linear and noncubane iron-sulfur clusters in EtfD resemble those in heterodisulfide reductases

The noncubane [4Fe-4S] and linear [3Fe-4S] clusters in EtfD are distinct from iron-sulfur clusters reported in other ETC complexes. A pair of noncubane [4Fe-4S] clusters was observed in the crystal structure of the cytoplasmic heterodisulfide reductase (HdrABC)–[-NiFe]-hydrogenase (MvhAGD) complex from the anaerobic methanogenic archaeaon *Methanothermococcus thermolithotrophicus* (Wagner et al, 2017; Watanabe et al, 2021). In HdrABC, the catalytic B and C subunits are homologous with the soluble region of EtfD, while subunit A acts as the link between the heterodisulfide reductase and the [NiFe] hydrogenase and is not homologous with EtfD (Fig. 3A, *middle*). The entire EtfD structure also overlays well with an AlphaFold model (Jumper et al, 2021) of the membrane-bound two subunit heterodisulfide reductase HdrDE, which is found in archaea such as *Methanosarcina barkeri* (Fig. 3A, *right*). The similarity of EtfD and the archaeal heterodisulfide reductases (Hdr) explains the presence of the noncubane [4Fe-4S] cluster in EtfD (Künkel et al, 1997; Wischgoll et al, 2005). In Hdr, the two noncubane iron-sulfur clusters are solvent-accessible and catalyze the homolytic cleavage of the heterodisulfide formed from coenzyme M and coenzyme B (CoM-S-S-CoB) (Fig. 3B, *top*). Each noncubane iron-sulfur cluster harbors a catalytic iron that is coordinated by a bridging cysteine (Cys81 and Cys234) that is displaced in the presence of the ligand (Fig. 3B, *bottom, red dashed lines*) (Pelmenschikov et al, 2023). This chemistry allows the

noncubane clusters to react with the disulfide bond in the substrate (Fig. 3B, *bottom, black dashed lines*) (Wagner et al, 2017), but is also expected to make the clusters in Hdr sensitive to oxygen (Imlay, 2006). In contrast, the iron-sulfur clusters D1 and D2 of EtfD are contained in a cavity that is shielded from the cytosol (Fig. 3C, *top*). In cluster D2, the noncubane geometry resembles the noncubane cluster in Hdr but its coordination does not rely on a bridging cysteine. Instead, a histidine at position 583 completes the coordination of the cluster, making it non-catalytic (Fig. 3C, *bottom right*). In cluster D1 of EtfD, a threonine at position 539 replaces the cysteine found in Hdr (Fig. 3C, *bottom left*). Therefore, EtfD cannot coordinate the fourth iron of a noncubane [4Fe-4S] cluster, and instead uses the cysteine at position 584 to coordinate the third iron and accommodate a linear [3Fe-4S] cluster (Fig. 3C, *bottom left*). The full coordination of the iron-sulfur clusters in EtfD, as well as their shielding from the outside solvent, may indicate adaptation of EtfD to function in aerobic conditions (Imlay, 2006).

## Development of a biochemical assay for mycobacterial EtfD activity

EtfD and its homologs were proposed to link β-oxidation to the ETC by accepting electrons from ETF and using them to reduce the membrane-bound menaquinone pool (Wischgoll et al, 2005; Agne et al, 2021; Beites et al, 2021) (Fig. 1A). This role was hypothesized, in part, because of the proximity of the gene encoding EtfD to the genes for the ETF subunits EtfA and EtfB in some bacterial genomes (Wischgoll et al, 2005). Immunoprecipitation confirmed the interaction of ETF with EtfD, with β-oxidation blocked at the acyl-CoA dehydrogenase step in a ΔetfD strain of *M. tuberculosis* (Beites et al, 2021). Based on this model, we developed an assay for EtfD activity (Fig. 4A) that resembles previously developed assays for mycobacterial ATP synthase activity, cyt. $bcc$-$aa_3$ activity, and mycobacterial NDH-2 activity (Harden et al, 2024; Liang et al, 2025). In this assay, a fatty acid substrate, acyl-CoA dehydrogenase (FadE) enzyme, ETF, mycobacterial inverted membrane vesicles (IMVs), and the fluorophore 9-amino-6-chloro-2-methoxyacridine (ACMA) are mixed in the wells of a plate in a fluorescence plate reader. The acyl-CoA substrate is oxidized by the FadE enzyme, which transfers electrons to ETF and subsequently EtfD in the IMV membrane. EtfD reduces the membrane-bound menaquinone pool, driving proton pumping by cyt. $bcc$-$aa_3$ and cyt. $bd$ and resulting in acidification of the IMV. This acidification causes fluorescence quenching of ACMA, which recovers when the ΔpH is collapsed by addition of the $K^+/H^+$ antiporter nigericin. Fluorescence quenching of 9-aminoacridines is proposed to result from interaction of the dye with the membrane, and increases on formation of a ΔpH (Elema et al, 1978; Huang et al, 1983; Grzesiek and Dencher, 1988; Grzesiek et al, 1989; Casadio, 1991; Casadio et al, 1995). Fluorescence quenching and recovery can be detected and quantified using the fluorescence plate reader. We selected butyryl-CoA as the substrate due to its solubility in aqueous solvent, and FadE5 as the acyl-CoA dehydrogenase because its ability to oxidize butyryl-CoA is well characterized (Chen et al, 2020). We prepared *M. smegmatis* strains with 3×FLAG tags at the C termini of the EtfA subunit of ETF and FadE5, and purified endogenous ETF and FadE5 from cultures of these strains (Fig. EV3A). *M. smegmatis* with no modification of EtfA, EtfB, or EtfD was used to generate wild-type IMVs for assays.

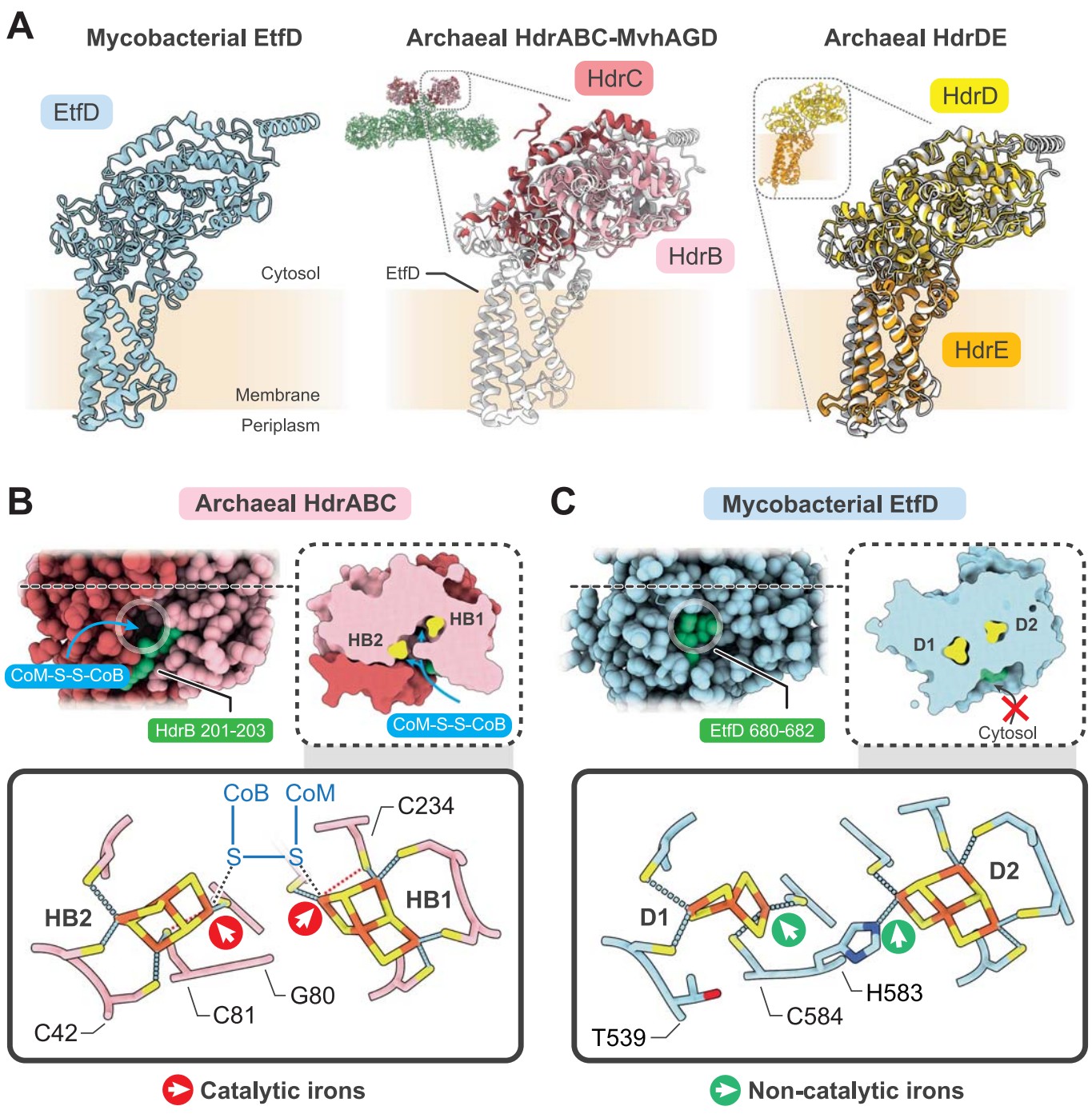

**Figure 3. Linear and noncubane iron-sulfur clusters are related to heterodisulfide reductase.**

(A) Comparison of the structure of EtfD (*left*), the archaeal heterodisulfide reductase HdrABC-MvhAGD (*middle*, PDB: 5ODH), and archaeal heterodisulfide reductase HdrDE (*right*, AF-P96796-F1, AF-P96797-F1). The homologous proteins EtfD, HdrBC, and HdrDE are overlayed. Full-length HdrABC-MvhAGD and HdrDE are shown as insets. Non-homologous subunits of HdrABC-MvhAGD are shown as green ribbons. (B) Location (*top*) and coordination (*bottom*) of the catalytic noncubane [4Fe-4S] clusters in HdrABC-MvhAGD. (C) Location (*top*) and coordination (*bottom*) of the corresponding linear [3Fe-4S] cluster D1 and noncubane [4Fe-4S] cluster D2 in EtfD. A stretch of equivalent residues from HdrB and EtfD based on multiple-sequence alignment is colored in green (*top*) to highlight the occlusion of the iron-sulfur cavity from the aqueous environment in EtfD.

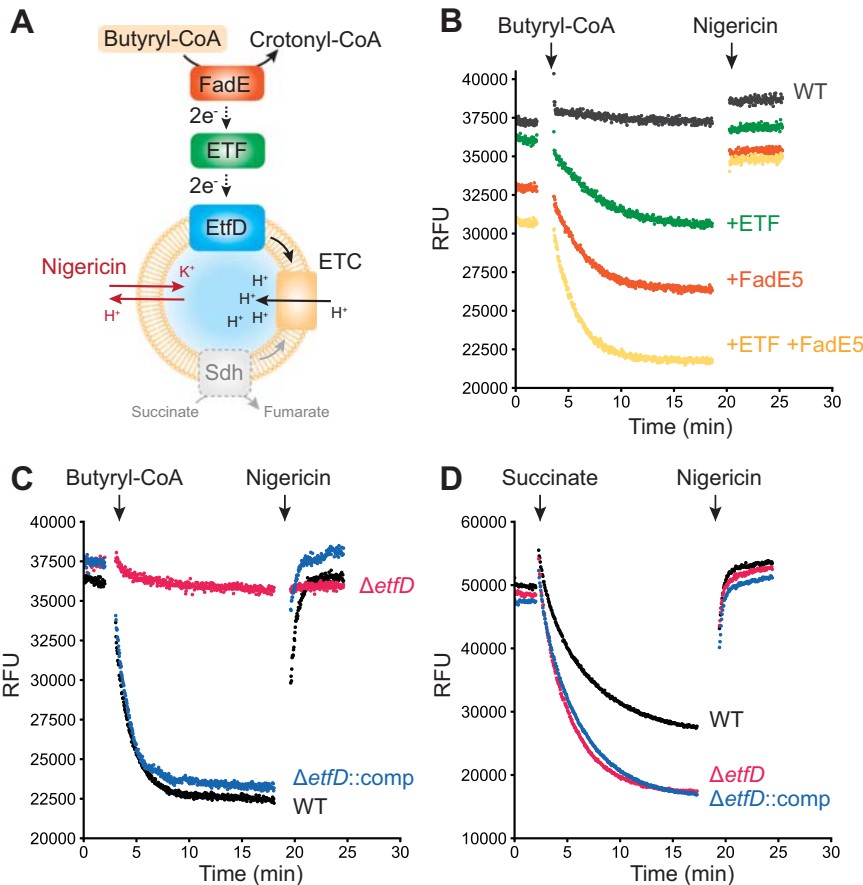

**Figure 4.  EtfD-mediated proton pumping in the presence of ETF and FadE5.**

(A) Schematic representation of an assay for EtfD activity. (B) Butyryl-CoA-driven proton pumping activity in IMVs from wild-type *M. smegmatis* (GMC_MSM1), supplemented with either buffer (*black*), ETF (*green*), FadE5 (*orange*), or ETF and FadE5 (*yellow*). (C) Butyryl-CoA-driven proton pumping activity in the presence of ETF and FadE5 for IMVs prepared from wild-type *M. smegmatis* (*black*), Δ*etfD M. smegmatis* (GMC_MSM9) (*red*), or Δ*etfD M. smegmatis* complemented with *M. tuberculosis* EtfD (GMC_MSM10) (*blue*). (D) Succinate-driven proton pumping activity in IMVs prepared from *M. smegmatis* that is either wild-type (*black*), Δ*etfD* (*red*), or Δ*etfD* complemented with *M. tuberculosis* EtfD (*blue*). Source data are available online for this figure.

Wild-type *M. smegmatis* IMVs do not display butyryl-CoA-dependent acidification (Fig. 4B, *black curve*), likely because little of the soluble proteins FadE and ETF co-purify with these vesicles. However, adding either ETF (Fig. 4B, *green*) or FadE5 (Fig. 4B, *orange*) leads to butyryl-CoA-dependent acidification of IMVs. The observation that adding either ETF or FadE5 allows for activity suggests that trace amounts of ETF and acyl-CoA dehydrogenases may contaminate the FadE5 and ETF protein preparations, respectively. The extent of acidification was increased by adding both ETF and FadE5 to the assay (Fig. 4B, *yellow*). Repeating the experiments that include ETF and FadE5, but using IMVs from a Δ*etfD M. smegmatis* strain shows that acidification is lost completely in the absence of EtfD (Fig. 4C, *red*). This acidification is rescued when IMVs are isolated from an *M. smegmatis* strain where the Δ*etfD* mutation is complemented with a plasmid encoding *M. tuberculosis* EtfD (Fig. 4C, *blue*). To ensure that the loss of activity of IMVs from the Δ*etfD* strain arises from a defect in EtfD activity, we confirmed that the IMVs remain capable of succinate-driven acidification, which relies on succinate dehydrogenase to drive proton pumping by the ETC (Fig. 4A) (Pecsi et al,

2014). IMVs from strains that are wild-type, Δ*etfD*, and Δ*etfD* complemented with a plasmid with *M. tuberculosis* EtfD, all display robust succinate-driven acidification (Fig. 4D). These results show that the proton pumping activity measured in the assay is dependent on EtfD, allowing for detection of EtfD activity. Replicates of the assays show the same results (Fig. EV3B–D).

## Effect of DBPI-11626157 on β-oxidation-driven IMV acidification

A series of 6,11-dioxobenzo[f]pyrido[1,2-a]indoles (DBPI) compounds have been shown to rely on EtfD to kill *M. tuberculosis* (Székely et al, 2020). To investigate the mechanism of action of one of these compounds, DBPI-11626157 (Fig. EV4A, *left*), we performed assays for butyryl-CoA-driven proton pumping using IMVs from the Δ*etfD* strain complemented with a plasmid for expression of *M. tuberculosis* EtfD. We used these IMVs because DBPIs are active against *M. tuberculosis* but not *M. smegmatis*. DBPI-11626157 at 10 μM (~6 times its MIC99 of 1.56 μM) inhibited butyryl-CoA-driven proton pumping by 44% (Fig. EV4B,D). To assess whether this inhibition was the result

of inhibiting electron transfer from β-oxidation to the ETC, we repeated the experiment but used succinate as the electron donor instead of butyryl-CoA. DBPI-11626157 at 10 µM similarly inhibited succinate-driven proton pumping by 46% (Fig. EV4C,D). Interestingly, we noticed that when DBPI-11626157 was present, fluorescence in the assay increased on addition of butyryl-CoA, even when IMV acidification was prevented by including nigericin at the beginning of the experiment (Fig. EV4E). However, this effect was also observed on addition of butyryl-CoA in the absence of ETF and FadE5, which suggests that it is unrelated to butyryl-CoA mediated-proton pumping (Fig. EV4F). This increase in fluorescence was not detected when succinate was used as the electron donor (Fig. EV4G). The reason for this effect is unclear but could be due to a small decrease in pH on addition of butyryl-CoA or an interaction between DBPI-11626157 and butyryl-CoA. Further, different concentrations of IMVs are used in the butyryl-CoA-driven and succinate-driven assays. This variation alters the DBPI:IMV ratio, which may affect the fluorescence. The equivalent inhibition of butyryl-CoA- and succinate-driven IMV acidification indicates that DBPI-11626157 is not a specific inhibitor of EtfD.

## Discussion

*M. tuberculosis* is thought to rely on lipid metabolism during persistence in granulomas (Russell et al, 2009; Wilburn et al, 2018), making EtfD an attractive target for drugs intended to reduce treatment duration. The experiments described here confirm the hypothesis that EtfD forms the electronic link between β-oxidation and the ETC and cryo-EM of EtfD demonstrates that it is structurally unrelated to the mammalian ETF-QO (Fig. EV5). We show that *M. smegmatis* provides a model for EtfD activity in mycobacteria, which may enable its use in phenotypic characterization of EtfD inhibitors. Further, we establish a biochemical assay for EtfD activity that can be used in a target-based screen for these compounds.

Disruption of EtfD activity prevents mycobacteria from metabolizing fatty acids and cholesterol as a source of energy, and leads to the accumulation of toxic intermediates in *M. tuberculosis* (Beites et al, 2021). Chemical inhibition of EtfD would not only block β-oxidation but also affect the ability of the ETC to establish a pmf. In *M. tuberculosis*, a Δ*etfD* strain can be complemented with Pox3 in vitro (Beites et al, 2021), which has acyl-CoA dehydrogenase activity analogous to FadEs but cannot transfer electrons to the ETC. This observation suggests that the energy contributed to the ETC from β-oxidation is not essential for *M. tuberculosis* survival in vitro, but β-oxidation itself, enabled by the oxidation of FadEs by ETF and EtfD, is required (Beites et al, 2021). Whether the contribution of EtfD to the pmf is essential or not in vivo remains to be investigated.

While the functional link between ETF and EtfD is clear (Wischgoll et al, 2005; Agne et al, 2021; Beites et al, 2021), ETF was not observed bound to endogenous EtfD in the structure of the protein presented here. This lack of binding could be because ETF shuttles electrons from numerous enzymes to EtfD (Henriques et al, 2021; Beites et al, 2021) and only interacts weakly with EtfD. A weak interaction, which could be lost during protein purification, would explain the lack of EtfD-mediated pumping activity by IMVs that are not supplemented with purified ETF. Alternatively, the structure determined here was obtained under oxidizing

conditions, and it is possible that only reduced ETF binds to EtfD. The disordered region of mycobacterial EtfD is not visible in the structure. The length of this region, which includes conserved interspersed repeats (Székely et al, 2020), differs between mycobacterial species (Fig. EV6). For example, *M. smegmatis* has a ~30 kDa disordered region, while in *Mycobacterium leprae* it is only ~13 kDa. An AlphaFold3-predicted structure of the ETF:EtfD complex suggests that ETF and EtfD interact through the disordered region of EtfD (Fig. EV7A–D).

In the absence of an ETF:EtfD structure, it is not clear which iron-sulfur cluster in EtfD accepts electrons from ETFs. The AlphaFold3 model places the FAD moiety of ETF ~12.5 Å from D1 and ~11.6 Å from D2 (Fig. EV7E), which are both sufficiently close for rapid electron transfer. Therefore, it is possible that both clusters are involved in mediating electron transfer from ETF to EtfD. The EtfD homolog FadF in *Bacillus subtilis* is known to be associated with β-oxidation (Matsuoka et al, 2007) and the *Pseudomonas aeruginosa* homolog DgcB participates in dimethylglycine catabolism (Wargo et al, 2008), a function that also involves ETF in mammals (Henriques et al, 2021). However, both FadF and DgcB lack the cysteine residues that coordinate cluster D1 (Fig. EV8). This observation suggests that FadF and DgcB also accept electrons from ETF and that electrons are transferred from the FAD in ETF directly to the equivalent of cluster D2. Therefore, D2 may also be the initial electron acceptor in EtfD. Interestingly, FadF and DgcB have shorter or entirely absent C-terminal disordered regions (<3 kDa).

The highly unusual linear [3Fe-4S] D1 cluster found in EtfD was first synthesized chemically and later detected spectroscopically in partially unfolded proteins and in the presence of ferrous ammonium sulfate or glutathione (Hagen and Holm, 1982; Kennedy et al, 1984; Gailer et al, 2001; Zhang et al, 2013). Consequently, these structures have been proposed to be degradation products or assembly intermediates of other iron-sulfur clusters (Krebs et al, 2000; Zhang et al, 2013). Linear [3Fe-4S] clusters were recently detected spectroscopically in overexpressed viral proteins under more physiological conditions (Villalta et al, 2023), but it is unknown how these clusters are stabilized in those proteins. The structure presented here demonstrates that a stable linear [3Fe-4S] cluster is found in a natively-folded protein and adopts a geometry similar to the one reported in the synthetic chemical system (Hagen and Holm, 1982). The similarity in coordination motifs and cluster geometry between the mycobacterial linear [3Fe-4S] cluster and the archaeal noncubane [4Fe-4S] cluster suggest that they are evolutionarily related, and that the linear cluster is unlikely to be a degradation product of a cubane [4Fe-4S] cluster. The linear [3Fe-4S] cluster binding motif can be found in EtfD homologs from other Actinobacteria (*Streptomyces griseus*) and in diverse phyla such as Acidobacteriota (e.g., *Acidobacterium capsulatum*), Deinococcota (e.g., *Deinococcus proteolyticus*), Spirochaetota (e.g., *Leptospirosa interrogans*), and Gemmatimonadota (e.g., *Gemmatirosa kalamazoonensis*) (Fig. EV8) (Agne et al, 2021).

In our assays, DBPI-11626157 shows similar inhibition of butyryl-CoA- and succinate-driven IMV acidification, suggesting that the compound does not inhibit EtfD specifically. DBPI-11626157 is a menaquinone derivative (Fig. EV4A, *right*), and therefore could interact with other enzymes that use menaquinone. DBPI-11626157 could also uncouple the pmf in IMVs non-

specifically, which could mask a weaker but specific inhibitory activity. The apparent collapse of the pmf in IMVs has been observed for a variety of compounds beyond traditional uncouplers, but is not well understood and does not necessarily translate to disruption of the pmf in live cells (Harrison et al, 2025; Fountain et al, 2025). Therefore, although DBPIs rely on EtfD to kill *M. tuberculosis* (Székely et al, 2020), their mode of action remains to be elucidated. Compounds that interact with the reporter dye, modulate protein interactions non-specifically, or are redox active could all interfere with the results of this assay. Nonetheless, the assay and structure presented here provide a framework for structure-guided drug discovery and development of direct-acting EtfD inhibitors.

# Methods

### Reagents and tools table

| Reagent/Resource | Reference or Source | Identifier or Catalog Number |
| --- | --- | --- |
| **Experimental models** | | |
| GMC_MSM1 | Rubinstein laboratory | |
| GMC_MSM4 | Rubinstein laboratory | |
| GMC_MSM5 | Rubinstein laboratory | |
| GMC_MSM9 | Rubinstein laboratory | |
| GMC_MSM10 | Rubinstein laboratory | |
| GMC_MSM11 | Rubinstein laboratory | |
| **Recombinant DNA** | | |
| pSAB41 | Rubinstein laboratory | |
| pKM444 | Addgene | 108319 |
| pKM464 | Addgene | 108322 |
| pGMCK-Ptb38-rv0338c | Schnappinger laboratory | |
| **Antibodies** | | |
| ANTI-FLAG M2 affinity gel | Millipore Sigma | A2220 |
| **Oligonucleotides and other sequence-based reagents** | | |
| ORBIT oligonucleotides | This study | Methods section |
| **Chemicals, Enzymes and other reagents** | | |
| DDM | Bluepus | DDM990 |
| Oleic acid | Millipore Sigma | 364525 |
| ACMA | Thermo Fisher Scientific | A1324 |
| Butyryl-CoA | Millipore Sigma | B1508 |
| Nigericin | Tocris Bioscience | 4312 |
| DBPI-11626157 | Makarov lab | |
| **Software** | | |
| cryoSPARC v.4 | https://cryosparc.com/ | |
| Clustal Omega | https://www.ebi.ac.uk/jdispatcher/msa/clustalo | |
| Jalview | https://www.jalview.org/ | |
| AlphaFold 3 | https://alphafoldserver.com/ | |

| Reagent/Resource | Reference or Source | Identifier or Catalog Number |
| --- | --- | --- |
| UCSF Chimera | https://www.cgl.ucsf.edu/chimera/ | |
| UCSF Chimera X | https://www.cgl.ucsf.edu/chimerax/ | |
| Coot | https://www2.mrc-lmb.cam.ac.uk/personal/pemsley/coot/ | |
| Phenix | https://www.phenix-online.org/ | |
| ISOLDE | https://tristanic.github.io/isolde/ | |
| **Other** | | |
| BioTek Synergy Neo2 Multi-mode Assay Microplate reader | Agilent Technologies | |
| ÄKTA pure | Cytiva | |
| Titan Krios G3 | Thermo Fisher Scientific | |
| EM GP2 plunge freezer | Leica Microsystems | |

### Strains

*M. smegmatis* strains GMC_MSM4 (EtfD-3×FLAG), GMC_MSM5 (EtfA-3×FLAG), GMC_MSM9 (Δ*etfD*) and GMC_MSM11 (FadE5-3×FLAG) were generated using the ORBIT method (Murphy et al, 2018) with the following oligonucleotides:

GCCCGAGAAGCCGGAACCGCCCGTGGTGGGGCTCGGCATCAAGCCGGGCGCCAAGCGGCCCGGTAAGCGCGGTTTGTCTGGTCAACCACCGCGGTCTCAGTGGTGTACGGTACAAACCTGACCGCGGCCACGCGTAGTTGGGAAGCGGCCCCCTCCTGATTCGGGAGGGGGCCGCTTCTTTCGGTAAG (GMC_MSM4),

AAGGGGTCCGGCAAAAAACACAGAACCCCGGCGCACCAGGTGCGCCGGGGTCCGCGTGATGAGGTGATCAGGTTTGTACCGTACACCACTGAGACCGCGGTGGTTGACCAGACAAACCGCCCTTGCGAGCCTTGACGGCCTCGGTGAGCTGCGGGCTGACCTTGAACAGGTCGCCCACGATGCCGAGG (GMC_MSM5),

TAAGTTACCGACGGGTAACGAACATATCGGGAGGCGCGCAGTGGCACACACCCTCGAAGTGAGCAGGCTCGGTTTGTCTGGTCAACCACCGCGGTCTCAGTGGTGTACGGTACAAACCCCGGGCGCCAAGCGGCCCGGTAAGCGCTGACCGCGGCCACGCGTAGTTGGGAAGCGGCCCCCTCCTGATT (GMC_MSM9),

GCTGACCAGCACGCGCCAGATCATCGAGAACCTCGA-CAACGACGTCATGGAGTTGGACGAGGCGGCGTTCGGTTTGTCTGGTCAACCACCGCGGTCTCAGTGGTGTACGGTACAAACCTAGAGGCGCGTTCTGAGGCAACGCTTCACCGAAGAGGCTCCCGGGTCACCCCGGGAGCCTCTTTGCATCT (GMC_MSM11). Strains with 3×FLAG tags were prepared using the payload plasmid pSAB41 (Guo et al, 2021) and the deletion strain was prepared using the payload plasmid pKM464 (Addgene #108322) (Murphy et al, 2018). A strain with Δ*etfD* complemented with *M. tuberculosis* EtfD, GMC_MSM10 (Δ*etfD::rv0338c*), was generated by transforming GMC_MSM9 with the previously-described integrating plasmid pGMCK-ptb38-*rv0338c* (Beites et al, 2021).

GMC_MSM1 (ATP synthase β subunit 3×FLAG) (Guo et al, 2021) was used as a wild-type strain for IMV acidification experiments.

## M. smegmatis growth assay

GMC_MSM4 (EtfD-3×FLAG), GMC_MSM9 (ΔetfD), GMC_MSM10 (ΔetfD::rv0338c) pre-cultures were grown in 7H9 medium supplemented with 0.5% (w/v) fatty acid-free bovine serum albumin (BSA), 0.08% (w/v) NaCl, 0.2% (w/v) D-glucose, 50 μg/mL hygromycin, and 0.05% (v/v) tyloxapol at 37 °C with shaking at 180 rpm. Log phase cultures ($OD_{600nm} = 0.4$ to 0.5) were centrifuged at $6500 \times g$ for 10 min and the bacteria were resuspended in phosphate buffered saline (PBS) to an $OD_{600nm}$ of 0.5. For growth assays, cultures were started at an $OD_{600nm}$ of 0.01 in 2 mL of modified Sauton medium (Beites et al, 2021) supplemented with 0.05% (v/v) tyloxapol, 0.5% (w/v) BSA, 0.08% (w/v) NaCl, 50 μg/mL hygromycin, and either 0.2% (w/v) D-glucose (~ 11 mM) or 250 μM oleic acid as the sole carbon source. Cultures grown with oleic acid were supplemented with an additional 250 μM oleic acid at the ~24 h and ~48 h timepoints to allow continued growth while avoiding acute toxicity, as described previously (Beites et al, 2021; Gouzy et al, 2021). Cultures were grown in 2 mL of medium in 14 mL polypropylene round-bottom tubes at 37 °C with shaking at 180 rpm. Turbidity was assessed by measuring $Abs_{580nm}$ of a 80 μL sample in a transparent flat-bottom 96-well microplate (Sarstedt) with a Spectramax M5e plate reader (Molecular Devices).

## Growth of M. smegmatis

For protein purification and IMV preparation, M. smegmatis strains were grown in 25 mL pre-cultures in Middlebrook 7H9 medium supplemented with 0.5% (w/v) BSA, 0.08% (w/v) NaCl, 0.2% (w/v) D-glucose, 50 μg/mL hygromycin, and 0.05% (v/v) Tween 80 at 37 °C with shaking at 180 rpm until saturation. From saturated cultures, 2 mL was used to inoculate each of six 1 L cultures grown in Middlebrook 7H9 medium supplemented with 0.08% (w/v) NaCl, 0.2% (w/v) D-glucose, 1% (w/v) tryptone, and 0.05% (v/v) Tween 80. Cells were grown for ~48 h at 30 °C with shaking at 180 rpm. All of the following steps were performed at 4 °C or on ice. Cells were collected by centrifugation at $6500 \times g$ for 20 min.

## Isolation of cytoplasmic and membrane fractions and preparation of IMVs

Cell pellets from 6 L cultures were resuspended in 150 mL Lysis buffer (50 mM Tris-HCl pH 7.5, 150 mM NaCl, 5 mM MgSO$_4$, 5 mM benzamidine hydrochloride, 5 mM aminocaproic acid, 1 mM PMSF). Cells were lysed at 20 kpsi with an Emulsiflex-C3 High-Pressure Homogenizer (Avestin) and then centrifuged at $39,000 \times g$ for 30 min. The supernatant was then centrifuged at $200,000 \times g$ for 1 h to collect membranes. For GMC_MSM5 (EtfA-3×FLAG) and GMC_MSM11 (FadE5-3×FLAG), the supernatant was collected to purify ETF and FadE5. For experiments with IMVs and for the purification of EtfD, the membrane fraction was resuspended in 15 mL (assays) or 40 mL (protein purification) S-buffer (50 mM Tris-HCl pH 7.5, 150 mM NaCl, 20% (v/v) glycerol, 5 mM MgSO$_4$, 5 mM benzamidine hydrochloride, 5 mM aminocaproic acid, 1 mM PMSF). This process results in formation of IMVs (Hertzberg and Hinkle, 1974; Rosen and Tsuchiya, 1979). IMVs were flash frozen

in liquid nitrogen, and stored at −80 °C. For IMV assays, resuspended membranes were aliquoted before freezing.

## IMV acidification assay

Proton pumping was assayed using IMVs harvested from GMC_MSM1, GMC_MSM9, and GMC_MSM10, with GMC_MSM1 acting as the wild-type control. Cells were grown for either 40 or 48 h. The assay was performed in IMV ACMA buffer (10 mM HEPES-KOH pH 7.5, 100 mM KCl, 5 mM MgCl$_2$, 3 μM ACMA) at a final volume of 160 μL. In this assay, the ACMA dye is added to the solution after IMVs are already formed. IMVs were used at a final concentration of ~1–1.5 mg/mL when butyryl-CoA was the electron donor, and diluted 16-fold when succinate was the electron donor. When comparing strains IMVs were diluted to normalize protein concentration. When used, ETF and FadE5 were added to IMVs at a final concentration of 2 μM each, or substituted with FLAG wash buffer 2 (50 mM Tris-HCl pH 7.4, 150 mM NaCl, 20% (v/v) glycerol, 5 mM MgSO$_4$, 5 mM benzamidine hydrochloride, 5 mM aminocaproic acid). In inhibition assays, DBPI-11626157 was first dissolved in DMSO at a concentration of 10 mM, stored at −20 °C and sonicated before use in a bath sonicator. The compound was added to the assay at a final DMSO concentration of 2%. The reaction was started with 2 mM (final concentration) butyryl-CoA lithium salt hydrate (MilliporeSigma) in MilliQ water (pH 4 to 5, measured with pH indicator strips), or 5 mM disodium succinate (pH ~8). To dissipate the proton gradient, nigericin (3.28 μL of a 50 μM stock in 1% ethanol) was added to the 160 μL reaction 15 min after the addition of the electron donor, reaching a final concentration of 1 μM. Black round-bottom 96-well microplates (BRANDTECH Scientific) were used, and the fluorescence was excited at 410 nm and monitored at 480 nm with a BioTek Synergy Neo2 Multi-mode Assay Microplate reader (Agilent Technologies). The difference in relative fluorescence units before and after addition of nigericin was calculated to quantify proton pumping activity. All values were divided by the mean fluorescence recovery of the DMSO controls to obtain relative activity. Importantly, identical assays performed with butyryl-CoA lithium salt hydrate from a different vendor (CHEMIMPEX) had a pH of ~7 when dissolved with MilliQ water and showed acidification of wild-type IMVs and ΔetfD IMVs both with and without ETF and FadE5, neither of which are expected to be capable of butyryl-CoA-driven acidification. This unexpected activity may be due to the presence of contaminants in the substrate that can donate electrons to other complexes of the ETC. Consequently, great care must be taken in identifying a suitable source of butyryl-CoA for the assay.

## Protein purification

To purify EtfD, membranes were solubilized in 1% (w/v) dodecyl maltoside (DDM) for 1 h. Insoluble material was removed by centrifugation at $200,000 \times g$ for 1 h. The supernatant was collected and filtered through a 0.45 μm filter before being applied to a column of 2 mL M2 affinity matrix (Sigma) previously equilibrated with FLAG wash buffer 1 (50 mM Tris-HCl pH 7.4, 150 mM NaCl, 15% (v/v) glycerol, 0.05% (w/v) DDM, 5 mM benzamidine hydrochloride, 5 mM aminocaproic acid). The column was then washed with ten column volumes of FLAG wash buffer 1, before protein was eluted with three column volumes of FLAG wash buffer 1 containing 150 μg/mL 3×FLAG peptide. The sample was then

concentrated with a 30 kDa cutoff concentrator (Sigma) to 500 µL before being loaded onto a Superose 6 Increase 10/300 gel filtration column (GE Healthcare) equilibrated in gel filtration buffer (50 mM Tris-HCl pH 7.4, 150 mM NaCl, 15% glycerol, 0.05% (w/v) DDM). Fractions containing EtfD were pooled and concentrated to ~12 µL with a 100 kDa cutoff concentrator (Sigma) for cryo-EM sample preparation.

To purify ETF and FadE5, the cytoplasmic fraction obtained after removing membranes from GMC_MSM5 and GMC_MSM11 cells was filtered with a 0.45 µM filter and applied to a column of 2 mL of M2 affinity matrix previously equilibrated using FLAG wash buffer 2 (50 mM Tris-HCl pH 7.4, 150 mM NaCl, 20% (v/v) glycerol, 5 mM MgSO$_4$, 5 mM benzamidine hydrochloride, 5 mM aminocaproic acid). The column was then washed with ten column volumes of FLAG wash buffer 2, before bound protein was eluted with three column volumes of FLAG wash buffer 2 containing 150 µg/mL 3×FLAG peptide. The eluted protein was then concentrated to ~30–60 µM with a 30 kDa cutoff concentrator.

## Cryo-EM sample preparation

Holey gold grids were made as described previously (Marr et al, 2014). Prior to grid freezing, glycerol was removed from the sample with a Zeba spin desalting column (Thermo Fischer) equilibrated in freeze buffer (50 mM Tris-HCl pH 7.4, 150 mM NaCl, 0.05% (w/v) DDM). Grids were glow discharged in air for 2 min and 2 µL of sample was applied to each grid in an EM GP2 plunge freezer (Leica) at 4 °C and 90% humidity. Grids were blotted for 1 s before freezing in liquid ethane.

## Data collection

Cryo-EM data was acquired with a 300 kV Titan Krios G3 electron microscope equipped with a Falcon 4i camera (Thermo Fischer Scientific). Data collection was automated with the EPU software package. 11,952 movies were collected in EER format (Guo et al, 2020) at 120,000× magnification, corresponding to calibrated pixel size of 0.64 Å. The exposure rate was 7.7 e⁻/pixel/s with a total exposure of ~70 e⁻/Å².

## Image analysis

CryoSPARC v.4 (Punjani et al, 2017) was used for image analysis. Movies were first aligned using patch motion correction and contrast transfer function (CTF) parameters were estimated in patches. Blob picking and 2D classification were used to generate 2D templates for template-based particle selection. Template selection yielded 1,588,703 particle images, which were extracted using a box size of 512 × 512 pixels, and Fourier cropped to 208 × 208 pixels. The dataset was curated using 2D classification and several rounds of ab-initio and heterogenous refinement, yielding 152,064 particle images. Particle images were then re-extracted using a box size of 512 × 512 pixels and Fourier cropped to 320 × 320 pixels. Non-uniform refinement (Punjani et al, 2020) and CTF refinement yielded a map of EtfD at 3.1 Å resolution. This resolution was improved to 2.9 Å with two rounds of reference-based motion correction and CTF refinement. Local refinement of the soluble region of EtfD yielded a map at 2.8 Å resolution. 3D classification was used to separate EtfD with an intact membrane

region from EtfD particles lacking the heme density. Non-uniform refinement of 48,099 intact particles from the 3D class that included the heme density yielded a 3.1 Å resolution map for the intact protein. However, non-uniform refinement of 48,053 particles images re-extracted from pre-processed movies resulted in a map that was at 3.2 Å resolution, but which showed better-defined map densities for the membrane-embedded region of EtfD.

## Atomic model building and refinement

An AlphaFold 3 (Jumper et al, 2021) prediction of the *M. smegmatis* EtfD structure (AFDB ID number: AF-A0QQB0-F1-v4) was fitted as a rigid body in the experimental density map for EtfD using UCSF Chimera (Goddard et al, 2007). The model was adjusted with Coot (Emsley and Cowtan, 2004) and refined with ISOLDE (Croll, 2018) and PHENIX (Afonine et al, 2018). Model statistics were evaluated with EMRinger (Barad et al, 2015) and Molprobity (Chen et al, 2010). Amino acid residues where the map lacked density for the side chain were truncated at the β carbon in the final model. Figures and movies were made using UCSF ChimeraX (Goddard et al, 2018) and Microsoft Clipchamp. Restraint files for the noncubane [4Fe-4S] and linear [3Fe-4S] clusters were generated with ELBOW (Moriarty et al, 2009) using the computationally-predicted structure of an open noncubane cluster (Pelmenschikov et al, 2023) and the crystal structure of the synthetic linear $[Fe_3S_4(SPh)_4]^{3-}$ cluster (Hagen and Holm, 1982).

## Sequence alignments

Sequences were aligned using Clustal Omega (Goujon et al, 2010; Sievers et al, 2011) and analyzed with Jalview (Waterhouse et al, 2009).

# Data availability

Movies are available on the Electron Microscopy Public Image Archive with the accession code EMPIAR-13058. Cryo-EM maps and the final atomic model are available through the Electron Microscopy Databank with accession codes EMD-70545 and EMD-70546, and Protein Databank with the accession code 9OJN. Bacterial strains are available from the authors.

The source data of this paper are collected in the following database record: biostudies:S-SCDT-10_1038-S44318-026-00726-y.

# Peer review information

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

## Acknowledgements

We thank Yingke Liang for advice on protein purification strategies. GMC was supported by a Mary H. Beatty Fellowship. JLR was supported by the Canada Research Chairs program. Research was funded by the Canadian Institutes of Health Research Project (grant PJT191893). SE and DS were supported by the National Institutes of Health (grant AI143575) and the Gates Foundation (grants INV-055896 and INV-055894). Cryo-EM data were collected at the Toronto High-Resolution High Throughput cryo-EM facility, supported by the Canada Foundation for Innovation and Ontario Research Fund. Assays were performed using infrastructure from the Structural and Biophysical Core Facility at The Hospital for Sick Children.

## Author contributions

**Gautier M Courbon**: Conceptualization; Data curation; Formal analysis; Validation; Investigation; Visualization; Methodology; Writing—original draft; Writing—review and editing. **Vadim Makarov**: Resources; Methodology. **Stewart T Cole**: Resources; Methodology. **Dirk Schnappinger**: Resources; Methodology. **Sabine Ehrt**: Resources; Methodology. **John L Rubinstein**: Resources; Formal analysis; Supervision; Funding acquisition; Visualization; Methodology; Writing—original draft; Project administration; Writing—review and editing.

In addition to the CRediT author contributions listed above, the contributions in detail are:

GMC conceived the project, designed and performed assays, purified protein, and collected and analyzed cryo-EM data. With JLR, GMC wrote the manuscript and prepared the figures with input from the other authors. VM and STC provided the compound DBPI-11626157 and advised on its use. DS and SE provided guidance on growth of *Mycobacterium smegmatis* with fatty acids as the energy source and provided the plasmid for expression of *M. tuberculosis* EtfD. JLR supervised the research, provided advice on the design of experiments, and coordinated the project. With GMC, JLR wrote the manuscript and prepared the figures with input from the other authors.

Source data underlying figure panels in this paper may have individual authorship assigned. Where available, figure panel/source data authorship is listed in the following database record: biostudies:S-SCDT-10_1038-S44318-026-00726-y.

## Disclosure and competing interests statement

The authors declare no competing interests.

# Expanded View Figures

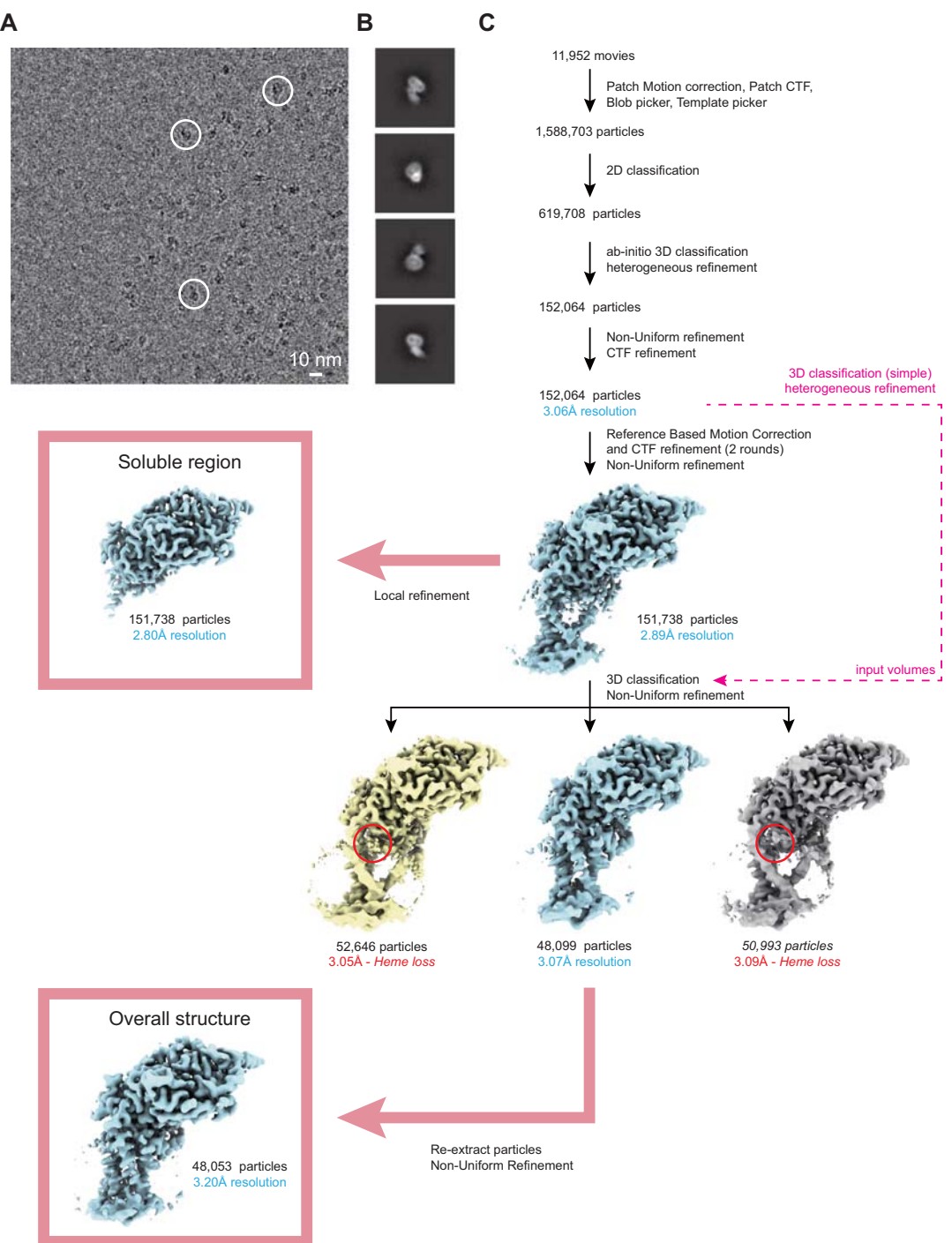

**Figure EV1.  Cryo-EM workflow.**

(**A**) Representative micrograph. Example particle images are circled. (**B**) 2D class averages of EtfD. (**C**) Simplified cryo-EM workflow. Red circles indicate the region in EtfD where the heme was lost.

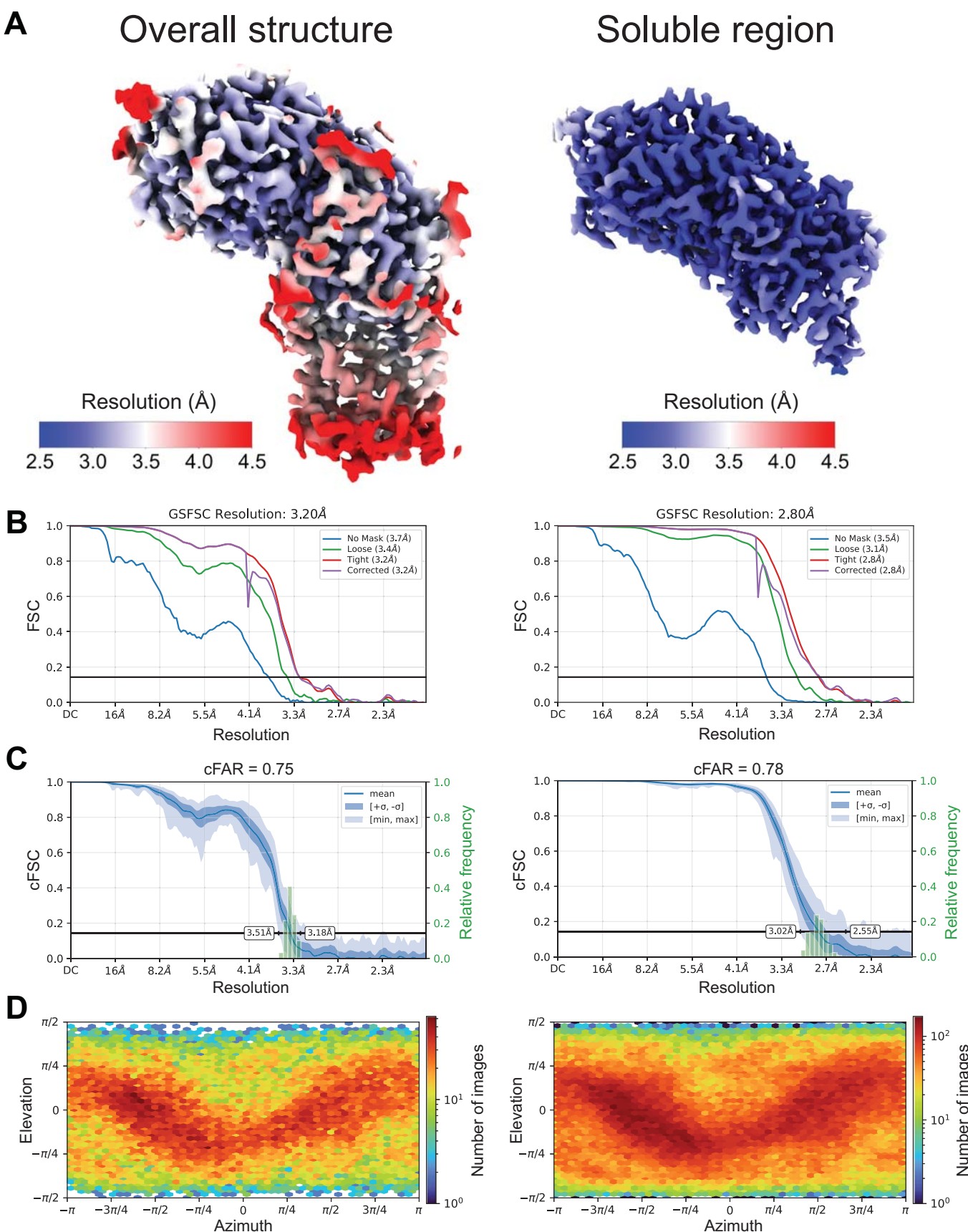

**Figure EV2.  Cryo-EM map validation.**

(**A**) Local resolution map for the full EtfD structure (*left*) and the locally refined soluble region of EtfD (*right*). (**B**) Fourier Shell Correlation (FSC) curves obtained after gold-standard refinement. (**C**) Conical FSC (cFSC) plots and their associated cFSC area ratio (cFAR) scores. (**D**) Particle orientation distribution plots.

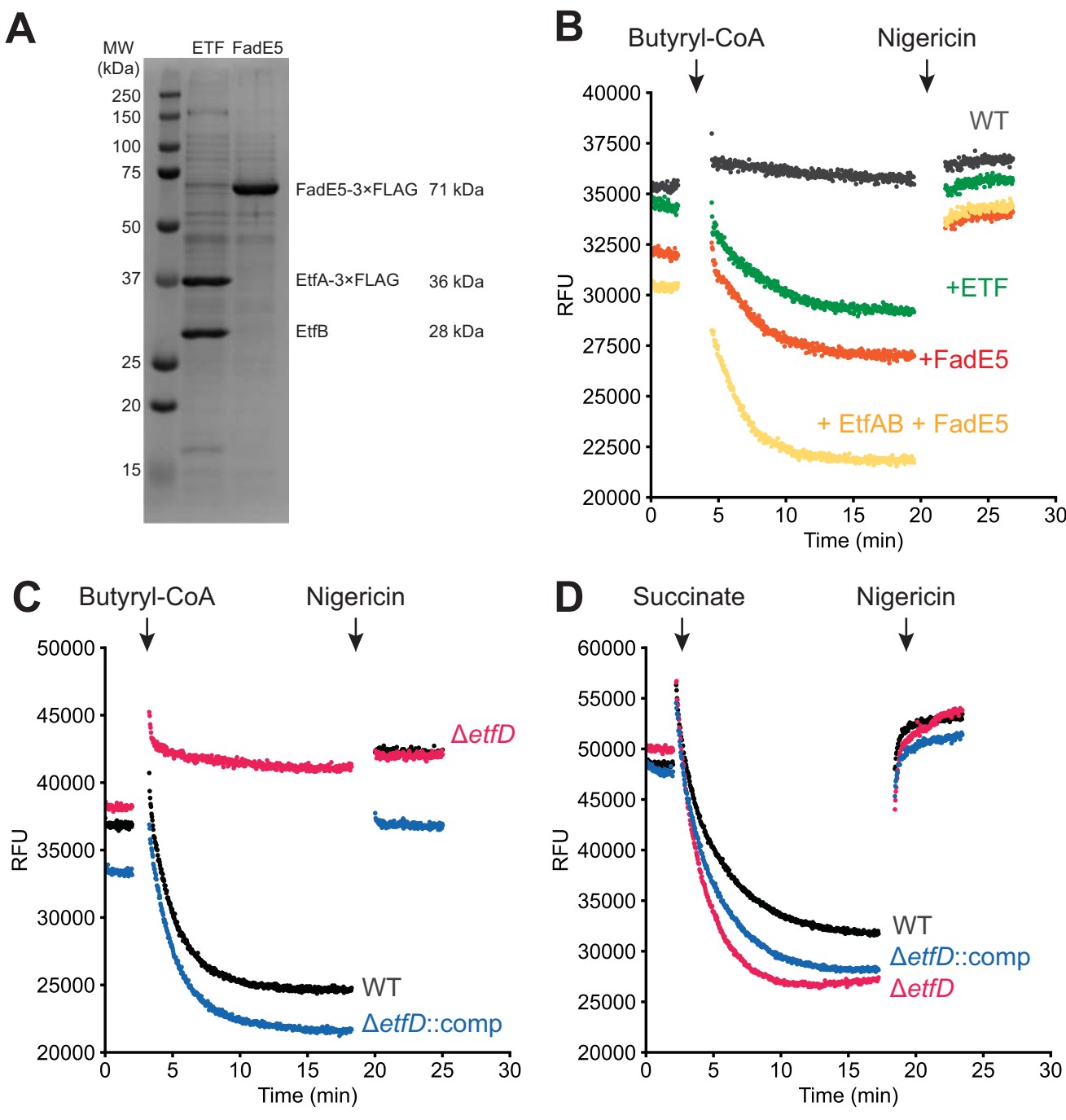

**Figure EV3. Assay replicates.**

(A) SDS-PAGE gels of purified ETF and FadE5. (B) Butyryl-CoA-driven proton pumping activity in IMVs prepared from wild-type *M. smegmatis* (GMC_MSM1) supplemented with either buffer (*black*), ETF (*green*), FadE5 (*orange*), or ETF and FadE5 (*yellow*). (C) Butyryl-CoA-driven proton pumping activity in the presence of ETF and FadE5 for IMVs prepared from *M. smegmatis* that is either wild-type (*black*), Δ*etfD* (*red*) (GMC_MSM9), or Δ*etfD* complemented with *M. tuberculosis* EtfD (GMC_MSM10). (D) Succinate-driven proton pumping activity in IMVs prepared from *M. smegmatis* that is either wild-type (*black*), Δ*etfD* (*red*), or Δ*etfD* complemented with *M. tuberculosis* EtfD (*blue*).

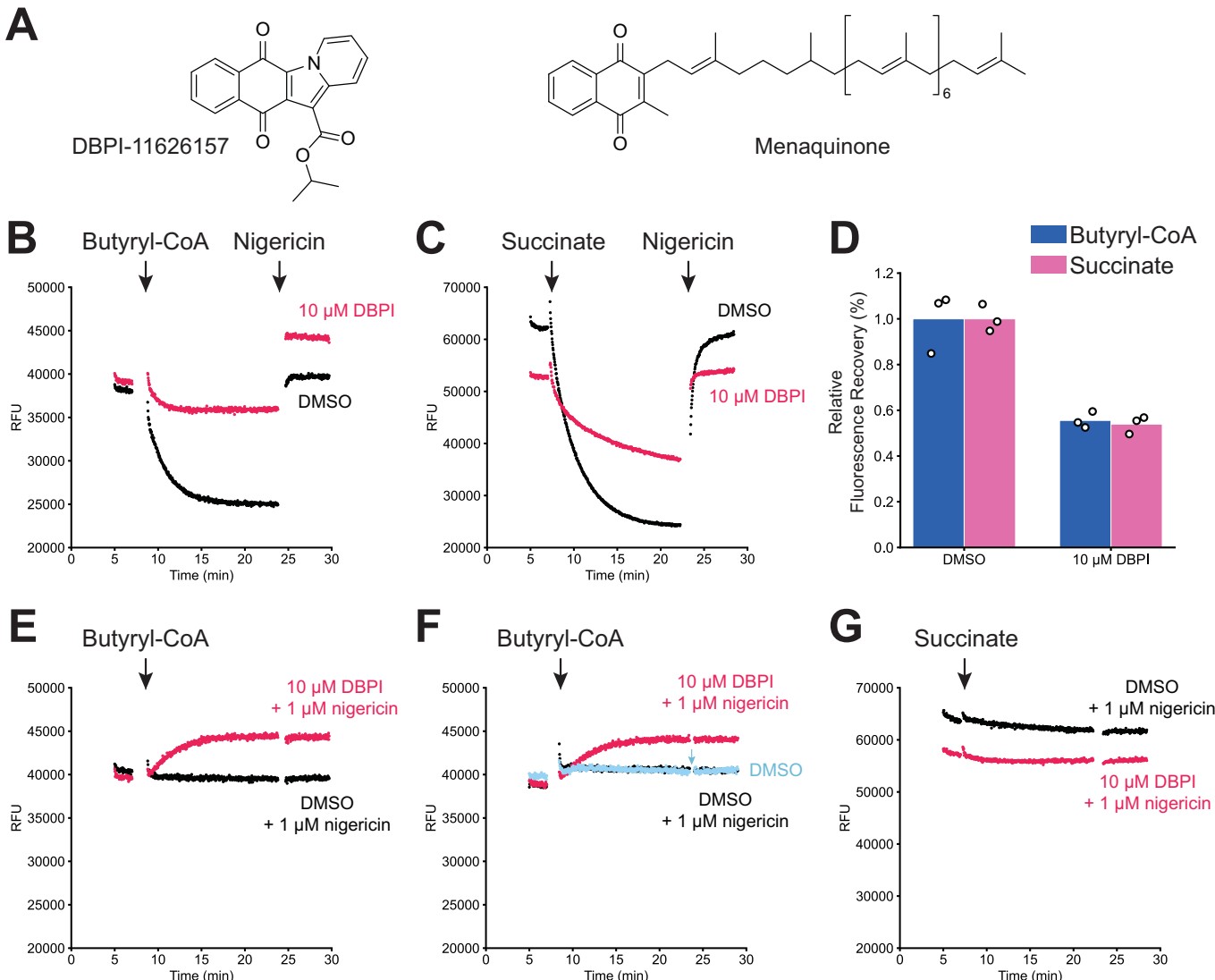

**Figure EV4.  Effect of DBPI-11626157 on butyryl-CoA-driven and succinate-driven IMV acidification.**

(A) Chemical structure of DBPI-11626157 (*left*) and menaquinone (*right*). (B) Butyryl-CoA-driven acidification of Δ*etfD::rv0338c* IMVs in the presence of 2 μM ETF and 1 μM FadE5, with 10 μM DBPI-11626157 (*red*) or DMSO (*black*). Data is representative of three replicates. (C) Succinate-driven acidification of Δ*etfD::rv0338c* IMVs with 10 μM DBPI-11626157 (*red*) or DMSO (*black*). Data is representative of three replicates. (D) Quantification of butyryl-CoA-driven and succinate-driven IMV acidification assays. (E) Addition of butyryl-CoA to Δ*etfD::rv0338c* IMVs in the presence of 1 μM nigericin, 2 μM ETF, and 1 μM FadE5, with 10 μM DBPI-11626157 (*red*) or DMSO (*black*). Data is representative of three replicates. (F) Addition of butyryl-CoA to Δ*etfD::rv0338c* IMVs without FadE5 or ETF with 10 μM DBPI-11626157 and 1 μM nigericin (*red*), DMSO and 1 μM nigericin (*black*), or DMSO alone (*blue*). Nigericin (1 μM) was added to the DMSO-only well during the assay (*blue arrow*). Data is representative of two replicates. (G) Addition of succinate to Δ*etfD::rv0338c* IMVs in the presence of 1 μM nigericin with 10 μM DBPI-11626157 (*red*) or 1 μM nigericin and DMSO (*black*). Data is representative of three replicates.

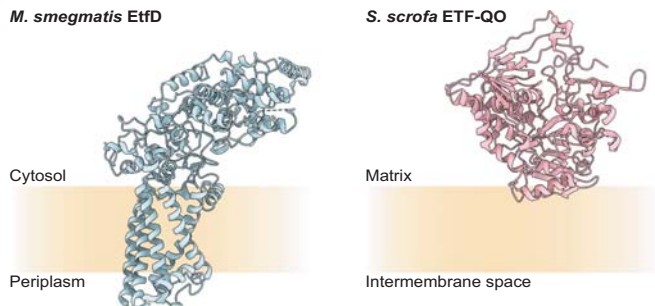

**M. smegmatis** EtfD

**S. scrofa** ETF-QO

Cytosol

Periplasm

Matrix

Intermembrane space

**Figure EV5.  Structure of mycobacterial and mammalian ETF dehydrogenases.**

Comparison of EtfD from *M. smegmatis* with ETF-QO oxidoreductase from *Sus scrofa* (PDB: 2GMH).

The EMBO Journal

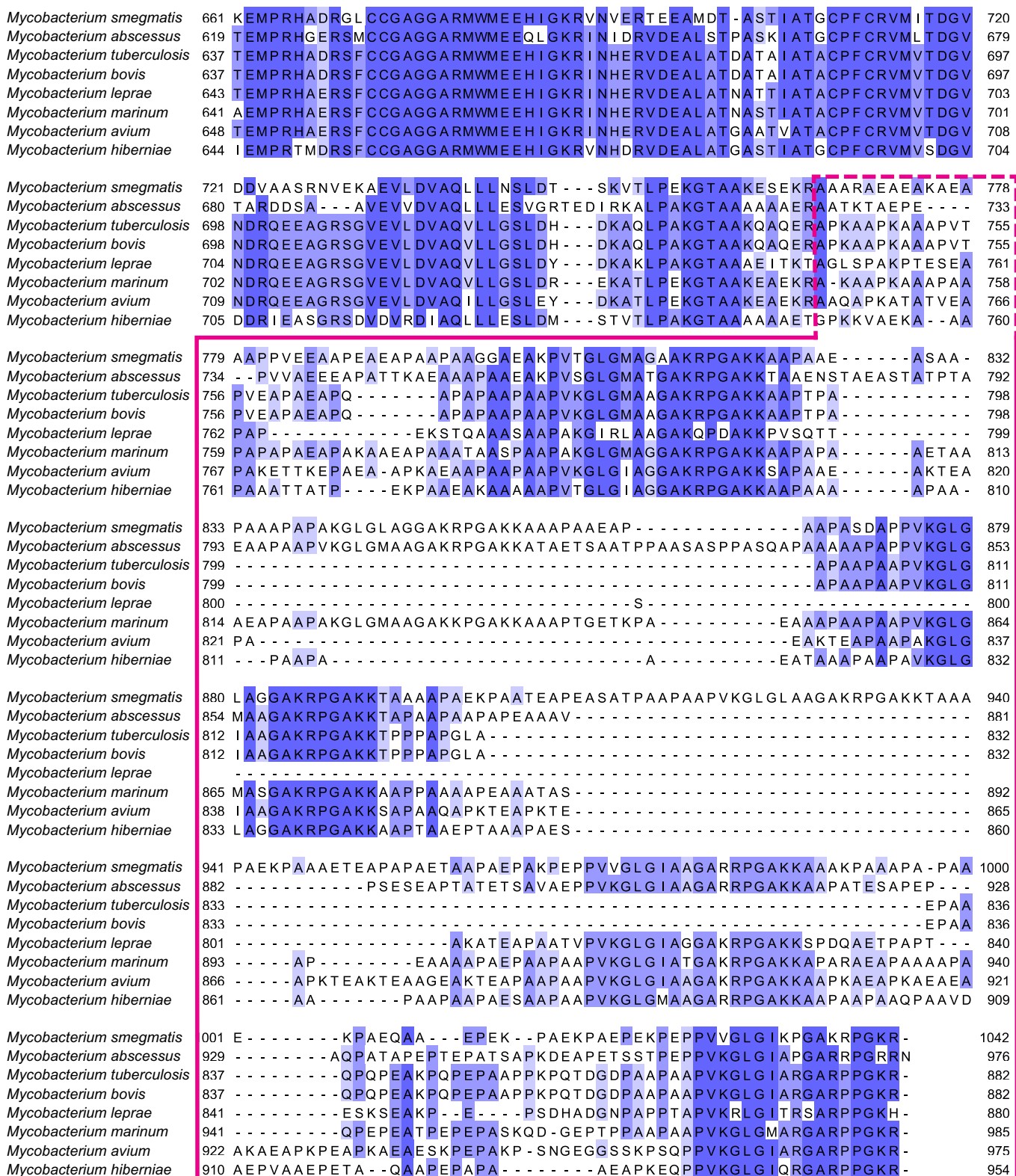

Disordered region

**Figure EV6.  Sequence alignment of the disordered region of mycobacterial EtfD homologs.**

Sequences are colored by conservation. The disordered region is indicated by a pink box. The start of the disordered region differs between species (dashed line).

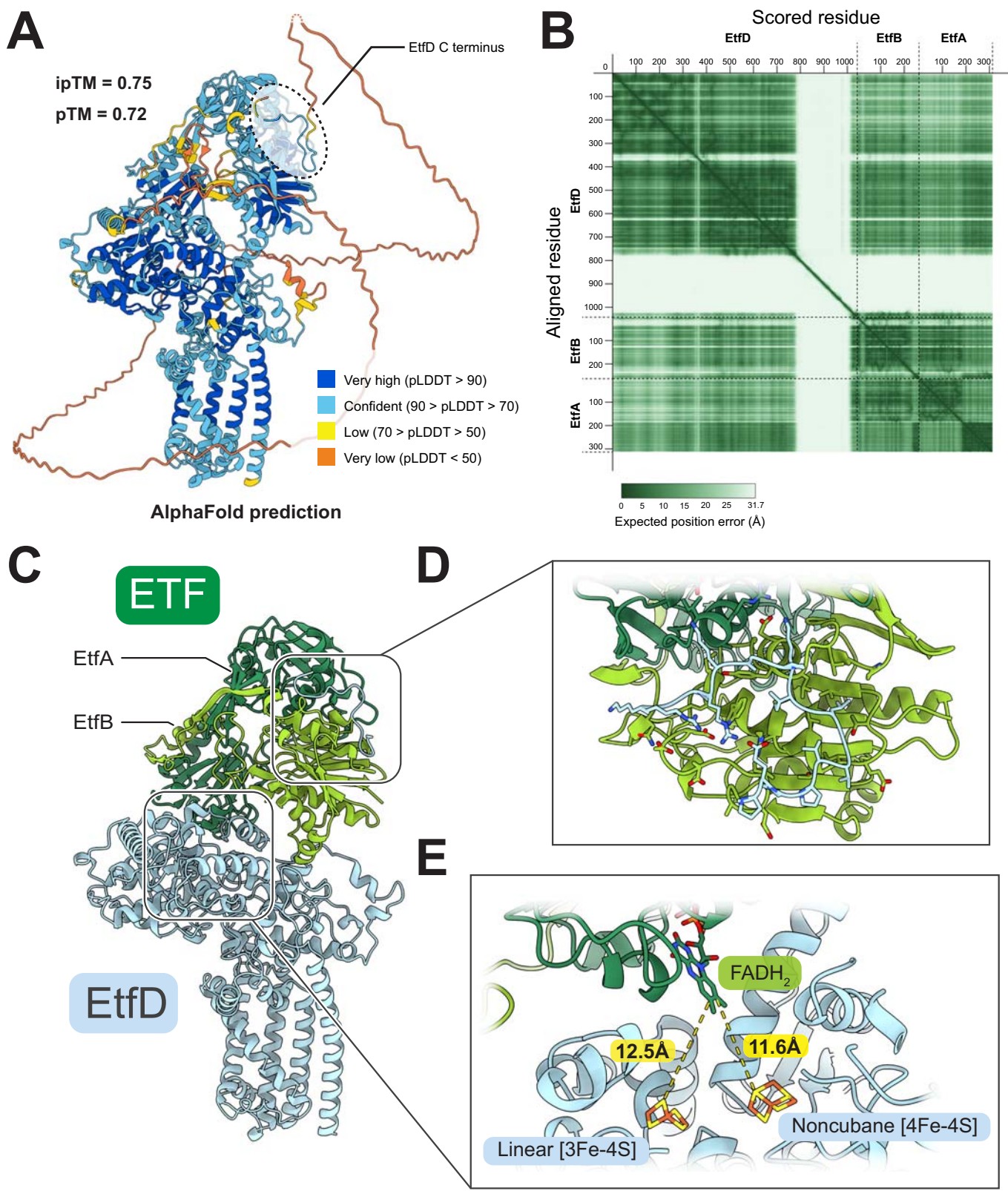

**Figure EV7.  AlphaFold prediction of the ETF:EtfD complex.**

(A) AlphaFold 3 prediction of the ETF-EtfD complex structure, colored by pLDDT score. (B) Predicted aligned error plot (Elfmann and Stülke, 2023). (C) Overview of the interaction between ETF (*green*) and EtfD (*blue*). The ~30 kDa disordered region of EtfD is not shown for clarity. (D) Close-up view of the interaction predicted by AlphaFold 3 between the disordered region of EtfD and ETF. (E) Location of the docked FAD cofactor of ETF from PDB 1EFV, relative to cluster D1 and D2 of EtfD.

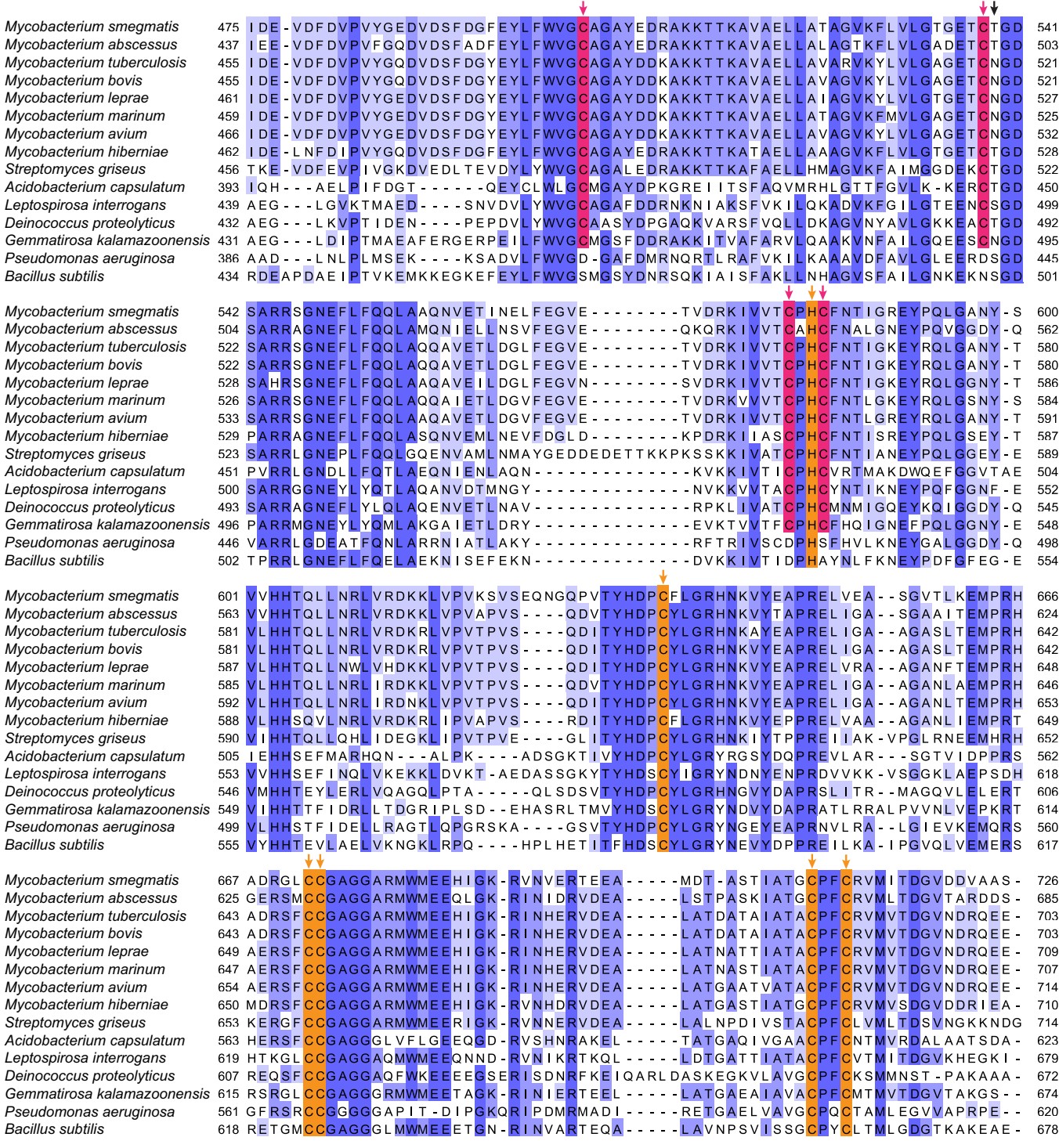

**linear [3Fe-4S]**    **noncubane [4Fe-4S]**

**Figure EV8.   Sequence alignment of the CCG domains of selected EtfD homologs.**

Coordinating residues are shown with red (linear [3Fe-4S] cluster) and orange (noncubane [4Fe-4S] cluster) arrows. The black arrow shows the position of the supernumerary cysteine in Hdr. Sequences are colored by conservation.

