## [Peer Review File · The EMBO Journal]

Structural basis for EtfD-mediated coupling of β -oxidation and the respiratory chain in mycobacteria

Gautier Courbon, Vadim Makarov, Stewart Cole, Dirk Schnappinger, Sabine Ehrt, and John Rubinstein

Corresponding author(s): John Rubinstein (john.rubinstein@utoronto.ca) , Gautier Courbon

(gautier.courbon@mail.utoronto.ca)

Review Timeline:

Submission Date:	2nd Jul 25
Editorial Decision:	19th Sep 25
Revision Received:	29th Oct 25
Editorial Decision:	11th Dec 25
Revision Received:	15th Dec 25
Accepted:	20th Jan 26

Editor: Cornelius Schneider

Transaction Report:

Dear Dr. Rubinstein,

Thank you for submitting your manuscript for consideration by the EMBO Journal. It has now been seen by two referees whose comments are shown below.

Given the referees' positive recommendations, I would like to invite you to submit a revised version of the manuscript, addressing the comments of all three reviewers. I should add that it is EMBO Journal policy to allow only a single round of revision, and acceptance of your manuscript will therefore depend on the completeness of your responses in this revised version.

Thank you for the opportunity to consider your work for publication. I look forward to your revision.

Yours sincerely,

Cornelius Schneider, PhD
Editor
The EMBO Journal
c.schneider@embojournal.org

Please remember: Digital image enhancement is acceptable practice, as long as it accurately represents the original data and conforms to community standards. If a figure has been subjected to significant electronic manipulation, this must be noted in the figure legend or in the 'Materials and Methods' section. The editors reserve the right to request original versions of figures and

the original images that were used to assemble the figure.

We realize that it is difficult to revise to a specific deadline. In the interest of protecting the conceptual advance provided by the work, we recommend a revision within 3 months (18th Dec 2025). Please discuss the revision progress ahead of this time with the editor if you require more time to complete the revisions. Use the link below to submit your revision:

Referee #1:

In this manuscript, Courbon et al. characterize EtfD from mycobacteria, which was proposed to link β -oxidation of fatty acids to the electron transport chain. Intriguingly, mycobacterial EtfD is not homologous to mammalian ETF-QO, suggesting that the mechanisms of electron shuttling from ETF to EtfD to menaquinone may be different. The authors determined the endogenous structure of EtfD from *M. smegmatis*, revealing an arrangement of redox cofactors (iron-sulfur clusters, heme, menaquinone) densities inside that delineate a possible electron transport path from ETF in the cytosol to the menaquinone pool in the membrane. The authors demonstrate that EtfD is needed for *M. smegmatis* growth in the presence of fatty acids as the sole carbon source and that EtfD from *M. tuberculosis* can complement an *M. smegmatis* etfD mutant, which may enable future efforts to screen for inhibitors of the Mtb enzyme.

Overall, this work is a significant advance in our understanding of metabolism in an important group of bacteria impacting human health. The structures appear to be well determined and refined, and the complementary biochemical/cellular experiments carried out with rigor and carefully interpreted. The structures are novel and potentially interesting to a broader community outside of mycobacterial research. I really enjoyed reading about the structural parallels between EtfD and the Hdr complexes, and how this fold and associated Fe-S clusters may have been repurposed through evolution for different functions in e- transfer vs disulfide reduction. The unusual, linear Fe-S cluster may also be of interest to many readers.

I have only relatively minor comments/suggestions to help the authors polish off this very interesting manuscript.

MAJOR COMMENTS

None.

MINOR COMMENTS

Lns 87-89: I suggest moving the sentence to the beginning of the inhibitor section (~Ln 253). I had forgotten this abbreviation by that point, and perhaps these drugs don't need to be introduced in the introduction (or could be introduced in a less detailed way if the authors still want to cite the reference there, e.g., "Series of compounds hypothesized to target EtfD have also been reported.").

Lns 152-158: I was initially concerned that the linear vs noncubane structures were based solely on the density maps (which are nearly sufficient quality to resolve this on their own, but perhaps not quite!). However, the authors discuss this in more detail in a subsequent section and I think the additional support from the coordinating residues and structurally related proteins provides nice support. I personally would have benefitted from a few words at this point along the lines of "The structure of these clusters will be discussed in more detail below" so I knew more info was coming and didn't start digging in to try to assess how the authors figured this out. Though this is a small stylistic detail that the authors are free to take or leave.

Lns 184-187: Is HdrBC also membrane associated? Since it wasn't mentioned, I am guessing that HdrA is not homologous to the TMD of EtfD or HdrE? I didn't initially notice the insets in Fig 3A, and they aren't mentioned in the legend. Perhaps some labels and/or addition to legend would be helpful. In particular, what is green in the panel labeled "Archaeal HdrABC-MvhAGD"? Is that just HdrA?

Lns 218-228, and also Fig 4a and Lns 499-527: Is ACMA trapped in the lumen of the IMVs, such that acidification of the ACMA-containing lumen leads to fluorescence quenching? This part of the assay is unclear to me, and the text and figure might benefit from clarification on this point (including how ACMA is loaded in the Methods, if appropriate).

Lns 246-247: It may be helpful to the reader to add a small comment to this sentence as to why succinate was used as substrate would be important to measure EtfD-dependent activity in this assay.

Lns 263-270: Do the authors have some hypothesis for why DBPI-11626157 seems to cause this change in fluorescence? Because I am not clear where ACMA is in this assays, I am having a little trouble rationalizing these results, though I agree that they seem to suggest that the target is something other than EtfD.

FigS1C: It may be helpful to the reader to indicate which pipeline is for the full-length map and which is for the soluble region. It also may be helpful to highlight the region in the 3D classification where the heme is present/ lost (perhaps by a circle).

Fig2C: Add in the corresponding colors to each redox cofactor in the legend.

FigS2B: It would be nice to see all the FSC curves, especially if a mask was used for local refinement of the soluble region.

Fig3B/C: What do the green spheres in the panels represent?

FigS5: Missing quality metrics for AF prediction. Add a panel showing PAE plot and perhaps complex colored by pLDDT score? It would be interesting to see the pLDDT score for the C-terminal regions proposed to interact with ETF.

There still isn't great consensus in the field about what should be in Table S1, but recently Gaber Lander proposed a nice list of stats in a thread on CCPEM (link below). I might suggest including some additional metrics to facilitate evaluation of data and structure quality. <https://www.jiscmail.ac.uk/cgi-bin/wa-jisc.exe?A2=ind2507&L=CCPEM&D=0&O=D&P=41630>

Would the authors consider depositing their raw micrographs in EMPIAR? This maximizes transparency/reproducibility/reuse of the data, with the added benefit that the data set is archived off site and they don't need to worry about maintaining a local copy themselves!

Referee #2:

Bacterial infections are responsible for the largest global numbers of death by infectious disease. More people are estimated to die yearly of bacterial infections than the total death-toll of COVID19. To invade human cells and to survive within them, bacteria depend on a reliable energy supply. Membrane proteins involved in energy metabolism are essential for the bacterial pathogenicity and they represent excellent targets for the development of new antimicrobials. An important premise is that these targets do not have structural homologues in humans that would render the drug potentially toxic. The protein EtfD coupling β -oxidation with respiration is one these promising targets as growth of *Mycobacterium tuberculosis*, the cause of tuberculosis, depends on this protein. However, as the structure of this protein is not known and its physiological role has not yet been proven, it is impossible to develop specific inhibitors against this protein.

In the manuscript, Courbon et al. work with a close relative of the pathogen, namely *Mycobacterium smegmatis*, a non-pathogenic and fast-growing model organism. They clearly show that *M. smegmatis* is also dependent on EtfD for growth on fatty acids. If the Krebs-cycle intermediate succinate is offered, the deletion of the corresponding gene has no effect. The authors isolate the protein and determine its structure by cryo-EM at a medium to good resolution. A clear model is obtained allowing the unambiguous assignment of the cofactors, namely four iron-sulfur clusters, one heme group as well as the substrate menaquinone. Surprisingly, the structure shows that one of the clusters is a so-called non-cubane [4Fe4S] cluster, while another is a linear [3Fe4S] cluster not yet discovered in native proteins. The non-cubane [4Fe4S] cluster is also found in heterodisulfide reductase (Hdr). The different functions of this cluster in both enzymes is very well discussed and explained. The structure and function of the linear [3Fe4S] cluster, previously known only as an inorganic model compound, is also convincingly described in great detail. The proposed electron path through EtfD and its interaction with its partner protein ETF as derived from AlphaFold predictions is highly convincing.

Furthermore, the authors report on the development of an assay to measure the activity of EtfD in vitro and thus offer a possibility for an HTS in the search for specific inhibitors. This is a coupled assay based on the oxidation of soluble fatty acids, whereby electrons are transferred to the quinol pool via three proteins (one of which is EtfD), thereby triggering proton translocation by the respiratory chain enzymes. A fluorimetric detection of this gradient is the read-out.

The manuscript is very well written, the text is concise and many data are provided. The quality of the structural data is excellent, therefore the structure can be considered as very reliable to justify the conclusions from this work. The finding of the non-cubane [4Fe4S] cluster is a big surprise and, together with the detection of the linear [3Fe4S] cluster, justifies the publication alone. All data are interpreted in a highly appropriate manner in the light of the current literature. The manuscript contributes new and most significant information about *Mycobacteria*, electron transfer and options for high-throughput screening. There are just a few points to be considered:

I have some problems with the proposed assay. Based on the controls shown, I think it is basically reasonable, but some details are still unclear to me. I do not understand why the addition of either ETF or FadE5 alone enables the reaction (Fig. 4). If, as mentioned in the text, trace amounts of each of these proteins were present in the test, a rapid reaction would take place immediately without further protein additions. The addition of ETF accelerates the reaction almost as clearly and quickly as that

of FadE5. A sufficient amount of FadE5 would therefore have to be present in the first experiment and a sufficient amount of ETF would have to be present in the second. However, the reaction would then have to take place without the added proteins.

Furthermore, I propose that the enhanced fluorescence found in the measurements with DBPI when using butyryl-CoA as substrate is due to a different protein plus lipid concentration in the assay. If I have understood the method section correctly, the suspension of IMVs is diluted 16-fold when succinate is used as a substrate (p. 25, l. 504). Thus, the (protein+lipid)/DBPI ratio differs significantly in both experiments. Due to the hydrophobicity of DBPI, micelles could form at low (protein/lipid) levels. This is already visible in the kinetics (Fig. S4): The starting value for the fluorescence measurements with butyryl-CoA is less than 40,000 RFU and for measurements with succinate it is about 60,000 RFU.

Finally, the assay is quite complex, and any compound that could interfere with protein/protein interactions will interfere with this assay. On the other hand, substances could also act as mediators and thus artificially accelerate electron transfer. The question therefore arises as to whether this is really a robust assay.

I like the discussion of electron transfer between ETF and EtfD very much (p. 10, l. 304 ff.). However, it could also be possible that the FAD of ETF donates one electron to cluster D1 and D2, each. Flavin may act as two electron donor, while each FeS cluster may accept one electron. Has anyone ever determined whether the FAD of ETF takes a radical state (indicating a transfer of a single electron)?

Minor remarks:

Fig. 2A: It would be great to label the clusters in the structure a bit more brilliant as they are hardly visible.

P. 26, l. 512: What is pH 4/5? I assume that a pH between 4 and 5 is meant, which should be described as such (pH 4 to 5).

We thank the editor and reviewers for their thorough evaluation and their enthusiasm for our manuscript. We are happy to report that we have addressed all the suggestions fully, and we believe the revised manuscript is now suitable for publication in *The EMBO Journal*.

Below, Reviewer comments are in blue with our responses in black.

Referee #1:

In this manuscript, Courbon et al. characterize EtfD from mycobacteria, which was proposed to link β -oxidation of fatty acids to the electron transport chain. Intriguingly, mycobacterial EtfD is not homologous to mammalian ETF-QO, suggesting that the mechanisms of electron shuttling from ETF to EtfD to menaquinone may be different. The authors determined the endogenous structure of EtfD from *M. smegmatis*, revealing an arrangement of redox cofactors (iron-sulfur clusters, heme, menaquinone) densities inside that delineate a possible electron transport path from ETF in the cytosol to the menaquinone pool in the membrane. The authors demonstrate that EtfD is needed for *M. smegmatis* growth in the presence of fatty acids as the sole carbon source and that EtfD from *M. tuberculosis* can complement an *M. smegmatis* etfD mutant, which may enable future efforts to screen for inhibitors of the Mtb enzyme.

Overall, this work is a significant advance in our understanding of metabolism in an important group of bacteria impacting human health. The structures appear to be well determined and refined, and the complementary biochemical/cellular experiments carried out with rigor and carefully interpreted. The structures are novel and potentially interesting to a broader community outside of mycobacterial research. I really enjoyed reading about the structural parallels between EtfD and the Hdr complexes, and how this fold and associated Fe-S clusters may have been repurposed through evolution for different functions in e- transfer vs disulfide reduction. The unusual, linear Fe-S cluster may also be of interest to many readers.

I have only relatively minor comments/suggestions to help the authors polish off this very interesting manuscript.

We thank the reviewer for their enthusiasm and kind comments.

MAJOR COMMENTS

None.

MINOR COMMENTS

Ln 87-89: I suggest moving the sentence to the beginning of the inhibitor section (~Ln 253). I had forgotten this abbreviation by that point, and perhaps these drugs don't need to be introduced in the introduction (or could be introduced in a less detailed way if the authors still want to cite the reference there, e.g., "Series of compounds hypothesized to target EtfD have also been reported.").

We thank the reviewer for this helpful suggestion. We have modified the text (Line 90):

"Further, a series of compounds have been shown to rely on EtfD to kill *M. tuberculosis* (Székely et al., 2020)"

and have moved the sentence with the abbreviation to Line 264:

“A series of 6,11-dioxobenzof[pyrido[1,2-a]indoles (DBPI) compounds have been shown to rely on EtfD to kill *M. tuberculosis* (Székely et al., 2020).”

Lns 152-158: I was initially concerned that the linear vs noncubane structures were based solely on the density maps (which are nearly sufficient quality to resolve this on their own, but perhaps not quite!). However, the authors discuss this in more detail in a subsequent section and I think the additional support from the coordinating residues and structurally related proteins provides nice support. I personally would have benefitted from a few words at this point along the lines of "The structure of these clusters will be discussed in more detail below" so I knew more info was coming and didn't start digging in to try to assess how the authors figured this out. Though this is a small stylistic detail that the authors are free to take or leave.

As suggested, we have added the sentence (Line 160):

“These unusual iron-sulfur clusters are discussed in detail in the next section.”

Lns 184-187: Is HdrBC also membrane associated? Since it wasn't mentioned, I am guessing that HdrA is not homologous to the TMD of EtfD or HdrE? I didn't initially notice the insets in Fig 3A, and they aren't mentioned in the legend. Perhaps some labels and/or addition to legend would be helpful. In particular, what is green in the panel labeled "Archaeal HdrABC-MvhAGD"? Is that just HdrA?

HdrBC is not membrane associated. The HdrABC-MvhAGD complex is a soluble protein composed of a heterodisulfide reductase (HdrABC) and an [NiFe] hydrogenase (MvhAGD). HdrA serves as the link between the HdrBC and mvhAGD subunits of the enzyme, and is not homologous with the transmembrane region of EtfD/HdrE. The green part of HdrABC-MvhAGD in the inset represent the subunits HdrA, MvhA, MvhG, and MvhD that are not homologous with EtfD.

We have updated the manuscript to clarify this point (Line 186):

“A pair of noncubane [4Fe-4S] clusters was observed in the crystal structure of the cytoplasmic heterodisulfide reductase (HdrABC)–[NiFe]–hydrogenase (MvhAGD) complex from the anaerobic methanogenic archaeon *Methanothermococcus thermolithotrophicus* (Wagner et al., 2017; Watanabe et al., 2021). In HdrABC, the catalytic B and C subunits are homologous with the soluble region of EtfD, while subunit A acts as the link between the heterodisulfide reductase and the [NiFe] hydrogenase and is not homologous with EtfD (**Fig. 3A, middle**).”

We have also updated figure legends to highlight the insets:

“**Figure 3. Linear and noncubane iron-sulfur clusters are related to heterodisulfide reductase.** **A**, Comparison of the structure of EtfD (*left*), the archaeal heterodisulfide reductase HdrABC-MvhAGD (*middle*, PDB: 5ODH), and archaeal heterodisulfide reductase HdrDE (*right*, AF-P96796-F1, AF-P96797-F1). The homologous proteins EtfD, HdrBC, and HdrDE are overlaid. Full length HdrABC-MvhAGD and HdrDE are shown as insets. Non-homologous subunits of HdrABC-MvhAGD are shown as green ribbons. **B**, Location (*top*) and coordination (*bottom*) of the catalytic noncubane [4Fe-4S] clusters in HdrABC-MvhAGD. **C**, Location (*top*) and coordination (*bottom*) of the corresponding linear [3Fe-4S] cluster D1 and noncubane [4Fe-4S] cluster D2 in EtfD. A stretch of

equivalent residues from HdrB and EtfD based on multiple-sequence alignment is colored in green (*top*) to highlight the occlusion of the iron-sulfur cavity from the aqueous environment in EtfD.”

Lines 218-228, and also Fig 4a and Lines 499-527: Is ACMA trapped in the lumen of the IMVs, such that acidification of the ACMA-containing lumen leads to fluorescence quenching? This part of the assay is unclear to me, and the text and figure might benefit from clarification on this point (including how ACMA is loaded in the Methods, if appropriate).

The molecular mechanism of ACMA and related dyes (such as 9-aminoacridine) as Δ pH indicators has been discussed in numerous papers but remains incompletely understood. In our experiments, ACMA is added to the bulk solution ahead of the assay and is not loaded into vesicles. ACMA is assumed to be membrane permeable in its neutral form, but membrane impermeable in its protonated form. It was originally postulated that upon establishing a Δ pH, 9-aminoacridine accumulates in the vesicle lumen, which causes fluorescence quenching (Schuldiner et al., 1972). However, it has also been shown that fluorescence quenching of 9-aminoacridines results from interaction of the dye with the membrane, which is enhanced upon formation of a Δ pH (Elema et al., 1978; Huang et al., 1983; Grzesiek and Dencher, 1988; Grzesiek et al., 1989). Some models combine elements from both theories (Casadio, 1991; Casadio et al., 1995).

We have modified the manuscript to highlight that ACMA is present in the bulk solution, and to introduce a proposed mechanism of action of ACMA (Line 524):

“In this assay, the ACMA dye is added to the solution after IMVs are already formed.”

And (Line 233):

“Fluorescence quenching of 9-aminoacridines is proposed to result from interaction of the dye with the membrane, and increases on formation of a Δ pH (Elema et al., 1978; Huang et al., 1983; Grzesiek and Dencher, 1988; Grzesiek et al., 1989; Casadio, 1991; Casadio et al., 1995).”

Lines 246-247: It may be helpful to the reader to add a small comment to this sentence as to why succinate was used as substrate would be important to measure EtfD-dependent activity in this assay.

We have modified the text to justify the succinate-driven acidification assay (Line 254):

“To ensure that the loss of activity of IMVs from the Δ etfD strain arises from a defect in EtfD activity, we confirmed that the IMVs remain capable of succinate-driven acidification, which relies on succinate dehydrogenase to drive proton pumping by the ETC (Fig. 4A) (Pecsi et al., 2014).”

Lines 263-270: Do the authors have some hypothesis for why DBPI-11626157 seems to cause this change in fluorescence? Because I am not clear where ACMA is in this assays, I am having a little trouble rationalizing these results, though I agree that they seem to suggest that the target is something other than EtfD.

The reason for change of fluorescence caused by the addition of butyryl-CoA is unclear. We initially hypothesized that the change in fluorescence may be due to the reduction of DBPI-

11626157 by EtfD, which could increase its fluorescence by reducing the naphthoquinone head to naphthoquinol. However, the change of fluorescence still occurs in the absence of EtfD, ETF, and FadE5, where no reduction is expected to occur. Perhaps the change in fluorescence could be due to a small decrease in pH on addition of the slightly acidic butyryl-CoA solution or an interaction between butyryl-CoA and DBPI.

We have modified the manuscript to highlight our uncertainty (Line 279):

“The reason for this effect is unclear but could be due to a small decrease in pH upon addition of butyryl-CoA or an interaction between DBPI-11626157 and butyryl-CoA.”

FigS1C: It may be helpful to the reader to indicate which pipeline is for the full-length map and which is for the soluble region. It also may be helpful to highlight the region in the 3D classification where the heme is present/ lost (perhaps by a circle).

We have modified Figure S1C according to the reviewer comment. The region in which the heme was lost is now indicated with a red circle. The map of the soluble region of EtfD and intact EtfD map have been annotated:

“**Figure S1. Cryo-EM workflow.** **A**, Representative micrograph. Example particle images are circled. **B**, 2D class averages of EtfD. **C**, Simplified cryo-EM workflow. Red circles indicate the region in EtfD where the heme was lost.”

Fig2C: Add in the corresponding colors to each redox cofactor in the legend.

We have updated the figure legend to add cofactor color:

“**Figure 2. Cryo-EM structure of *M. smegmatis* EtfD.** **A**, Cryo-EM map (*left*) and atomic model (*right*) of EtfD. DDM, dodecyl maltoside. **B**, Schematic representation of EtfD domain organization. Domains were annotated using Pfam (Finn et al., 2008). **C**, Arrangement of redox cofactors in EtfD. Iron-sulfur clusters are shown in yellow and orange, heme *b* in pink, and menaquinone in green. All cofactors are shown as space filling models. A possible

electron path from ETF to menaquinone is shown by the blue arrow and the edge-to-edge distance between redox co-factors is indicated. **D**, Coordination of the redox cofactors in EtfD. The cryo-EM map density is shown in blue.”

FigS2B: It would be nice to see all the FSC curves, especially if a mask was used for local refinement of the soluble region.

The figure has been updated to show all FSC curves as requested by the reviewer.

Fig3B/C: What do the green spheres in the panels represent?

The green spheres represent equivalent regions of archaeal HdrABC and mycobacterial EtfD based on multiple sequence alignment. It highlights that these residues occlude the iron-sulfur cavity in EtfD. The figure caption has been updated to clarify this point:

“**Figure 3. Linear and noncubane iron-sulfur clusters are related to heterodisulfide reductase.** **A**, Comparison of the structure of EtfD (*left*), the archaeal heterodisulfide reductase HdrABC-MvhAGD (*middle*, PDB: 5ODH), and archaeal heterodisulfide reductase HdrDE (*right*, AF-P96796-F1, AF-P96797-F1). The homologous proteins EtfD, HdrBC and HdrDE are overlaid. Full length HdrABC-MvhAGD and HdrDE are shown as insets. Non-homologous subunits of HdrABC-MvhAGD are shown as green ribbons. **B**, Location (*top*) and coordination (*bottom*) of the catalytic noncubane [4Fe-4S] clusters in HdrABC-MvhAGD. **C**, Location (*top*) and coordination (*bottom*) of the

corresponding linear [3Fe-4S] cluster D1 and noncubane [4Fe-4S] cluster D2 in EtfD. A stretch of equivalent residues from HdrB and EtfD based on multiple-sequence alignment is colored in green (*top*) to highlight the occlusion of the iron-sulfur cavity from the aqueous environment in EtfD.”

FigS5: Missing quality metrics for AF prediction. Add a panel showing PAE plot and perhaps complex colored by pLDDT score? It would be interesting to see the pLDDT score for the C-terminal regions proposed to interact with ETF.

We thank the reviewer for this suggestion. We have modified the figure to include a panel showing the PAE plot and the structure colored by pLDDT score. The C-terminal region interacting with ETF has a pLDDT score between 70 and 90.

The figure caption was modified as follows:

“**Figure S5. AlphaFold prediction of the ETF:EtfD complex.** **A**, AlphaFold 3 prediction of the ETF-EtfD complex structure, coloured by pLDDT score. **B**, Predicted aligned error plot (Elfmann and Stülke, 2023). **C**, Overview of the interaction between ETF (*green*) and EtfD (*blue*). The ~30 kDa disordered region of EtfD is not shown for clarity. **D**, Close-up view of the interaction predicted by AlphaFold 3 between the disordered region of EtfD and ETF. **E**, Location of the docked FAD cofactor of ETF from PDB 1EFV, relative to cluster D1 and D2 of EtfD.”

There still isn't great consensus in the field about what should be in Table S1, but recently Gaber Lander proposed a nice list of stats in a thread on CCPEM (link below). I might suggest including some additional metrics to facilitate evaluation of data and structure quality. <https://www.jiscmail.ac.uk/cgi-bin/wa-jisc.exe?A2=ind2507&L=CCPEM&D=0&O=D&P=41630>

We have added the following metrics (highlighted in green) to table S1 as suggested by the reviewer:

	EtfD (EMD-70545, PDB 9OJN)	EtfD – soluble region (EMD-70546)
Data collection and processing		
Microscope		Titan Krios G3
Detector		Falcon 4i
Automation software		EPU
Magnification		120,000
Voltage (kV)		300
Electron exposure (e ⁻ /Å ²)		70
Exposure rate (e ⁻ /pixel/s)		7.7
Exposure time (s)		3.7
Defocus range (µm)		0.8-2.3
Pixel size (Å)		0.64
Movies used for processing (no.)		11,952
Symmetry imposed		C1
Initial particle images (no.)		619,708
Final particle images (no.)	48,053	151,738
Map resolution (Å)		
FSC threshold	0.143/0.5	0.143/0.5
Masked	3.2/3.5	2.8/3.2
Unmasked	3.7/7.3	3.5/7.4
Map resolution range (Å)	2.9-5.0	2.5-3.1
Refinement		
Initial model used (accession code)	AF-A0QQB0-F1-v4	
Model resolution (Å)	3.3	
FSC threshold	0.5	
Map sharpening B factor (Å ²)	-65.6	
Model composition		
Non-hydrogen atoms	5,601	
Protein residues	742	
Ligands	1 LMT 1 MQ9 1 HEM 2 SF4	

	1 A1CBX 1 9S8	
R.m.s deviations		
Bond lengths (Å)	0.004	
Bond angles (°)	0.954	
Validation		
MolProbity score	1.22	
Clashscore	4.40	
Rotamer outliers (%)	0.00	
CaBLAM outliers (%)	0.68	
EMRinger score	4.48	
Average Q-score	0.528	
CC (volume)	0.78	
CC (mask)	0.80	
Ramachandran plot		
Favored (%)	98.1	
Allowed (%)	1.90	
Outliers (%)	0.00	

Would the authors consider depositing their raw micrographs in EMPIAR? This maximizes transparency/reproducibility/reuse of the data, with the added benefit that the data set is archived off site and they don't need to worry about maintaining a local copy themselves!

We thank the author for this suggestion. The raw micrographs are now accessible on EMPIAR. The accession code has been added to the Data availability section:

“Movies are available on the Electron Microscopy Public Image Archive with the accession code EMPIAR-13058.”

Referee #2:

Bacterial infections are responsible for the largest global numbers of death by infectious disease. More people are estimated to die yearly of bacterial infections than the total death-toll of COVID19. To invade human cells and to survive within them, bacteria depend on a reliable energy supply. Membrane proteins involved in energy metabolism are essential for the bacterial pathogenicity and they represent excellent targets for the development of new antimicrobials. An important premise is that these targets do not have structural homologues in humans that would render the drug potentially toxic. The protein EtfD coupling β -oxidation with respiration is one these promising targets as growth of *Mycobacterium tuberculosis*, the cause of tuberculosis, depends on this protein. However, as the structure of this protein is not known and its physiological role has not yet been proven, it is impossible to develop specific inhibitors against this protein.

In the manuscript, Courbon et al. work with a close relative of the pathogen, namely *Mycobacterium smegmatis*, a non-pathogenic and fast-growing model organism. They clearly show that *M. smegmatis* is also dependent on EtfD for growth on fatty acids. If the Krebs-cycle intermediate succinate is offered, the deletion of the corresponding gene has no effect. The authors isolate the protein and determine its structure by cryo-EM at a medium to good resolution. A clear model is obtained allowing the unambiguous assignment of the cofactors, namely four iron-sulfur clusters, one heme group as well as the substrate menaquinone. Surprisingly, the structure shows that one of the clusters is a so-called non-cubane [4Fe4S]

cluster, while another is a linear [3Fe4S] cluster not yet discovered in native proteins. The non-cubane [4Fe4S] cluster is also found in heterodisulfide reductase (Hdr). The different functions of this cluster in both enzymes is very well discussed and explained. The structure and function of the linear [3Fe4S] cluster, previously known only as an inorganic model compound, is also convincingly described in great detail. The proposed electron path through EtfD and its interaction with its partner protein ETF as derived from AlphaFold predictions is highly convincing.

Furthermore, the authors report on the development of an assay to measure the activity of EtfD in vitro and thus offer a possibility for an HTS in the search for specific inhibitors. This is a coupled assay based on the oxidation of soluble fatty acids, whereby electrons are transferred to the quinol pool via three proteins (one of which is EtfD), thereby triggering proton translocation by the respiratory chain enzymes. A fluorimetric detection of this gradient is the read-out.

The manuscript is very well written, the text is concise and many data are provided. The quality of the structural data is excellent, therefore the structure can be considered as very reliable to justify the conclusions from this work. The finding of the non-cubane [4Fe4S] cluster is a big surprise and, together with the detection of the linear [3Fe4S] cluster, justifies the publication alone. All data are interpreted in a highly appropriate manner in the light of the current literature. The manuscript contributes new and most significant information about Mycobacteria, electron transfer and options for high-throughput screening. There are just a few points to be considered:

We thank the reviewer for their kind comments and we are delighted to hear their enthusiasm about the unusual iron-sulfur clusters observed in our study.

I have some problems with the proposed assay. Based on the controls shown, I think it is basically reasonable, but some details are still unclear to me. I do not understand why the addition of either ETF or FadE5 alone enables the reaction (Fig. 4). If, as mentioned in the text, trace amounts of each of these proteins were present in the test, a rapid reaction would take place immediately without further protein additions. The addition of ETF accelerates the reaction almost as clearly and quickly as that of FadE5. A sufficient amount of FadE5 would therefore have to be present in the first experiment and a sufficient amount of ETF would have to be present in the second. However, the reaction would then have to take place without the added proteins.

We thank the reviewer for their careful review of the assay. We apologize for not being sufficiently clear.

We agree with the reviewer that trace amounts of ETF and FadE5 are likely not present in the membrane fraction collected to form IMVs, which explains the lack of reaction in the black WT trace in Fig. 4B. However, ETF and acyl-CoA dehydrogenases are interacting proteins. We hypothesize that some amount of ETF co-purifies with FadE5, and similarly some acyl-CoA dehydrogenases (such as FadE5) co-purify with ETF. Consequently, addition of either purified endogenous *M. smegmatis* ETF or FadE5 to wild-type IMVs introduces some of the other enzyme, which explains the pumping observed in the green and orange curves in Fig. 4B. We have modified the manuscript to clarify this hypothesis (Line 247):

“The observation that adding either ETF or FadE5 allows for activity suggests that trace amounts of ETF and acyl-CoA dehydrogenases may contaminate the FadE5 and ETF protein preparations, respectively.”

Furthermore, I propose that the enhanced fluorescence found in the measurements with DBPI when using butyryl-CoA as substrate is due to a different protein plus lipid concentration in the assay. If I have understood the method section correctly, the suspension of IMVs is diluted 16-fold when succinate is used as a substrate (p. 25, l. 504). Thus, the (protein+lipid)/DBPI ratio differs significantly in both experiments. Due to the hydrophobicity of DBPI, micelles could form at low (protein/lipid) levels. This is already visible in the kinetics (Fig. S4): The starting value for the fluorescence measurements with butyryl-CoA is less than 40,000 RFU and for measurements with succinate it is about 60,000 RFU.

We thank the reviewer for their insight. We have observed that the starting fluorescence in the assay decreases with higher concentrations of IMVs. Therefore, it is expected that for an identical gain the succinate condition (1/16 [IMV]) would start at a higher base fluorescence than the butyryl-CoA condition (1/1 [IMV]). As the reviewer points out, this would also lead to a different DBPI/IMV ratio in the two experiments, which may affect the fluorescence reading. We have modified the manuscript to highlight the reviewer’s observation (Line 281):

“Further, different concentrations of IMVs are used in the butyryl-CoA-driven and succinate-driven assays. This variation alters the DBPI:IMV ratio, which may affect the fluorescence.”

Finally, the assay is quite complex, and any compound that could interfere with protein/protein interactions will interfere with this assay. On the other hand, substances could also act as mediators and thus artificially accelerate electron transfer. The question therefore arises as to whether this is really a robust assay.

We understand the reviewer’s concern about the complexity of the assay. While this type of assay has been used successfully for characterization of specific inhibitors that target the mycobacterial oxidative phosphorylation machinery (Harden et al., 2024; Liang et al., 2025), and in unpublished experiments, successful detection of new inhibitors, we agree with the reviewer that some compounds could interfere with the results. We have modified the manuscript to highlight these limitations (Line 360):

“Compounds that interact with the reporter dye, modulate protein interactions non-specifically, or are redox active could all interfere with the results of this assay.”

I like the discussion of electron transfer between ETF and EtfD very much (p. 10, l. 304 ff.). However, it could also be possible that the FAD of ETF donates one electron to cluster D1 and D2, each. Flavin may act as two electron donor, while each FeS cluster may accept one electron. Has anyone ever determined whether the FAD of ETF takes a radical state (indicating a transfer of a single electron)?

The reviewer raises an excellent point. After some investigation, we found that studies have determined that the FAD of ETF can exist in a radical state (Hall and Lambeth, 1980; Gorelick et al., 1985). These studies suggest that electron transfer between acyl-CoA dehydrogenases and

ETF happens in two consecutive one-electron steps requiring two ETFs. If this mechanism of action is conserved in mycobacteria, ETF would donate electrons to EtfD one at a time to reduce menaquinone to menaquinol. How this two-step process is catalyzed by EtfD remains unclear. We have modified the manuscript to mention the results of these studies (Line 65):

The FAD cofactor reduced in β -oxidation is found in a FadE protein, an acyl-CoA dehydrogenase, which transfers electrons to a FAD-containing Electron Transfer Flavoprotein (ETF). Although FAD can be reduced by two electrons to form FADH₂, ETF has been proposed to oxidize acyl-CoA dehydrogenases in two successive one-electron steps requiring two ETFs (Hall and Lambeth, 1980; Gorelick et al., 1985)."

We have also modified the text to mention the possibility that both clusters participate in electron transfer from ETF to EtfD (Line 320):

"The AlphaFold3 model places the FAD moiety of ETF ~12.5 Å from D1 and ~11.6 Å from D2 (**Fig. S5D**), which are both sufficiently close for rapid electron transfer. Therefore, it is possible that both clusters are involved in mediating electron transfer from ETF to EtfD."

Minor remarks:

Fig. 2A: It would be great to label the clusters in the structure a bit more brilliant as they are hardly visible.

The figure has been updated to make the redox cofactors more visible.

P. 26, l. 512: What is pH 4/5? I assume that a pH between 4 and 5 is meant, which should be described as such (pH 4 to 5).

The text has been edited to explicitly state that the pH of butyryl CoA solution was between 4 and 5. The reason for this uncertainty was that the volume for this expensive reagent was extremely small and consequently the pH could only be measured with pH indicator strips (Line 533):

"The reaction was started with 2 mM (final concentration) butyryl-CoA lithium salt hydrate (MilliporeSigma) in MilliQ water (pH 4 to 5, measured with pH indicator strips), or 5 mM disodium succinate (pH ~8)."

Dear Dr. Rubinstein,

Thank you for submitting a revised version of your manuscript. Your study has now been seen by all original referees, who find that their previous concerns have been addressed and now recommend publication of the manuscript. There remain only a few mainly editorial points that have to be addressed before I can extend formal acceptance of the manuscript:

- 1) Please remove all figures from the ms with legends placed below the References, please also remove track changes
- 2) AFFILIATIONS (research institution or university vs. biotech company): employment in a biotech company should be stated in DCIS
- 3) Please reduce the number of keywords on the abstract page to five (ideally choosing broad general terms).
- 4) Please remove the DOI numbers from references
- 5) Please rename the Conflict of Interest section into "Disclosure and Competing Interests Statement", in accordance with our updated Guide to Authors (<https://link.springer.com/partners/embo-press/editorial-policies#Competing%20interest%20disclosures>)
- 6) As we are switching from a free-text author contribution statement towards a more formal statement based on Contributor Role Taxonomy (CRediT) terms, please remove the present Author Contribution section and instead specify each author's contribution(s) directly in the Author Information page of our submission system during upload of the final manuscript. See <https://casrai.org/credit/> for more information.
- 7) FIGURE CALLOUTS: There are missing callouts for panels of EV figures
- 8) Please provide the Reagent and Tools Table. For more information, please check <https://media.springernature.com/original/springer-cms/rest/v1/content/27825802/data/v1>
- 9) SOURCE DATA: Source data files need to be saved in a scheme one figure/folder and then uploaded as .zip files. E.g. all the Source data files for figure 1 need to be saved in a single folder and this needs to be zipped and then uploaded as "SD figure 1.zip" file. For EV and/or appendix figures, ZIP together all source data. Completed SD checklist should be uploaded as Related Manuscript File.
- 10) Please provide suggestions for a short 'blurb' text prefacing and summing up the conceptual aspect of the study in two sentences (max. 250 characters), followed by 3-5 one-sentence 'bullet points' with brief factual statements of key results of the paper; they will form the basis of an editor-written 'Synopsis' accompanying the online version of the article. Please also provide an altered synopsis image, making sure that the aspect ratio conforms to our website's format - it should be exactly 550 pixels wide and between 300-600 pixels high.
- 11) Please note that the specific URLs for EMPIAR-13058, EMD-70545 and EMD-70546 datasets are not provided in the data availability statement.
- 12) Table EV1 should be uploaded as an individual file or renamed to Table 1 with the corresponding callout and placed between main and EV figure legends
- 13) Movie file: The legend should be removed from ms and zipped with the movie file.
- 14) Sections need to be named and the order should be corrected: Title page - Abstract - Keywords - Introduction - Results - Discussion - Methods - Data Availability - Acknowledgements - Disclosure and Competing Interests Statement - References - Figure Legends - Table(s) - Expanded View Figure Legends.

With best regards,
Cornelius Schneider

Cornelius Schneider, PhD
Editor | The EMBO Journal
c.schneider@embojournal.org

Please refer to our figure preparation guideline in order to ensure proper formatting and readability in print as well as on screen:

<https://link.springer.com/journal/44318/submission-guidelines#cms-Figure-and-data-presentation>

Referee #1:

The authors have address all my comments. A great manuscript!

Referee #2:

The authors have taken all of the reviewers' concerns into account and amended the text of the manuscript accordingly. The revised version of the manuscript can therefore be accepted without reservation.

All minor editorial requests have been addressed by the authors.

Dear Dr. Rubinstein,

I am pleased to inform you that your manuscript has been accepted for publication in the EMBO Journal.

You may qualify for financial assistance for your publication charges - either via a Springer Nature fully open access agreement or an EMBO initiative. Check your eligibility: <https://link.springer.com/journal/44318/how-to-publish-with-us>

Yours sincerely,

Cornelius Schneider, PhD
Editor
The EMBO Journal
c.schneider@embojournal.org

Please note that it is The EMBO Journal policy for the transcript of the editorial process (containing referee reports and your response letters) to be published as an online supplement to each paper. If you should prefer removal of any referee-only figures included in the point-by-point response(s), e.g. because they may still be used for future publication or because they have been reproduced from published work by others, please do let us know immediately via response email.

More information is available here: <https://link.springer.com/partners/embo-press/editorial-policies#Peer%20review>